# On the Efficacy of Server-Aided Federated Learning against Partial Client Participation

## Abstract

Although federated learning (FL) has become a prevailing distributed learning framework in recent years due to its benefits in scalability/privacy, there remain many significant challenges in FL system design. Notably, most existing works in the current FL literature assume either full client or uniformly distributed client participation. Unfortunately, this idealistic assumption rarely holds in practice. It has been frequently observed that some clients may never participate in FL training (aka partial/incomplete participation) due to a meld of system heterogeneity factors. To mitigate impacts of partial client participation, an increasingly popular approach in practical FL systems is the sever-aided federated learning (SA-FL) framework, where one equips the server with an auxiliary dataset. However, despite the fact that SA-FL has been empirically shown to be effective in addressing the partial client participation problem, there remains a lack of theoretical understanding for SA-FL. Worse yet, even the ramifications of partial worker participation are not clearly understood in conventional FL so far. These theoretical gaps motivate us to rigorously investigate SA-FL. To this end, we first reveal that conventional FL is *not* PAC-learnable under partial client participation in the worst case, which advances our understanding of conventional FL. Then, we show that the PAC-learnability of FL with partial client participation can indeed be revived by SA-FL, which theoretically justifies the use of SA-FL for the first time. Lastly, to further make SA-FL communication-efficient, we propose the SAFARI (server-aided federated averaging) algorithm that enjoys convergence guarantees and the same level of communication efficiency and privacy as the state-of-the-art FL.

## 1 Introduction

Since the seminal work by McMahan et al. (2017), federated learning (FL) has emerged as a powerful distributed learning paradigm that enables a large number of clients (e.g., edge devices) to collaboratively train a model under a central server's coordination. However, with FL gaining popularity, it has also become apparent that FL faces a key challenge that is unseen in traditional distributed learning in datacenter settings – *system heterogeneity*. Generally speaking, system heterogeneity in FL is caused by the massively different computation and communication capabilities at each client (computational power, communication capacity, drop-out rate, etc.). Studies have shown that system heterogeneity can significantly impact client participation in a highly non-trivial fashion and severely degrade the learning performance of FL algorithms (Bonawitz et al., 2019; Yang et al., 2021a). For example, it is shown in (Yang et al., 2021a) that more than $30\%$ clients never participate in FL, while only $30\%$ of the clients contribute to $81\%$ of the total computation even if the server uniformly samples the clients. Exacerbating the problem is the fact that the client's status could be unstable and time-varying due to the aforementioned computation and communication constraints.

To mitigate the impact of partial client participation, one approach called *server-aided federated learning* (SA-FL) has been increasingly adopted in practical FL systems in recent years (see, e.g., (Zhao et al., 2018; Wang et al., 2021b)). The basic idea of SA-FL is to equip the server in FL with a small auxiliary dataset that approximately mimics the population distribution, so that the distribution deviation induced by partial client participation can be corrected. To date, even though SA-FL has been empirically shown to be quite effective in addressing the partial client participation problem in practice, there remains *a lack of theoretical understanding* for SA-FL. This motivates us to rigorously investigate the efficacy of SA-FL against partial client participation in FL in this paper.

Somewhat counterintuitively, to fully understand SA-FL, one must be able to first see what happens if SA-FL is not used and partial client participation is left untreated in conventional FL. In other words, we need to first answer the following fundamental question: *"1) What are the impacts of partial client participation on FL learning performance?"* Upon answering this question, the next important follow-up question regarding SA-FL is: *"2) What benefits could SA-FL bring and how could we theoretically characterize them?* Also, since SA-FL still largely follows the server-client architecture that demands intensive communication between server and clients, the third fundamental question regarding SA-FL is: *"3) Could we make SA-FL as communication-efficient as conventional FL?"* Indeed, answering these three questions constitutes the rest of this paper, where we address the first two questions through the lens of PAC (probably approximately correct) learnability, while resolving the third question by proposing a communication-efficient SA-FL algorithm. Our major contributions in this work are summarized as follows:

- By establishing a *worst-case* generalization error lower bound, we show that conventional FL is *not PAC-learnable* under partial client participation. In other words, no learning algorithm can approach zero generalization error under partial participation for conventional FL even in the limit of infinitely many data samples and training iterations. This insight, though being negative, warrants the necessity of developing new algorithmic techniques and system architectures (e.g., SA-FL) to modify the conventional FL framework to mitigate partial client participation.

- Inspired by techniques from domain adaptation, we prove a new generalization error bound to show that SA-FL can indeed *revive the PAC learnability of FL* under partial client participation. We note that this bound could reach zero asymptotically as the number data samples increase. This is much stronger than previous results in domain adaptation with non-vanishing small error (see Section 2 for details).

- To make SA-FL communication-efficient, we propose a new training algorithm for SA-FL called SAFARI (server-aided federated averaging). By carefully designing the update coordination between the server and the clients, we show that SAFARI achieves an $\mathcal{O}(1/\sqrt{KR})$ convergence rate, matching the convergence rate of state-of-the-art conventional FL algorithms. We also conduct extensive experiments to demonstrate the efficacy and efficiency of our SAFARI algorithm.

The rest of this paper is organized as follows. In Section 2, we review the literature to put our work in comparative perspectives. Section 3 presents the PAC learning analysis of standard FL under partial participation and our proposed SA-FL framework. We then propose SAFARI algorithm with convergence guarantees in Section 4.

## 2  RELATED WORK

**1) Client Participation in Federated Learning:** The seminal FedAvg algorithm was first proposed in (McMahan et al., 2017) as a heuristic to improve communication efficiency and data privacy for FL. Since then, there have been many follow-ups (e.g., (Li et al., 2020a; Wang et al., 2020; Zhang et al., 2020; Acar et al., 2021; Karimireddy et al., 2020; Luo et al., 2021; Mitra et al., 2021; Karimireddy et al., 2021; Khanduri et al., 2021; Murata & Suzuki, 2021; Avdiukhin & Kasiviswanathan, 2021) and so on) on addressing the data heterogeneity challenge in FL. However, most of these works (e.g., (Li et al., 2020a; Wang et al., 2020; Zhang et al., 2020; Acar et al., 2021; Karimireddy et al., 2020; Yang et al., 2021b)) are based on the full or uniform (i.e., sampling clients uniformly at random) client participation assumption. The full or uniform participation assumptions are essential since they are required to ensure that the stochastic gradient estimator is unbiased in each round of update. Thus, even if "model drift" or "objective inconsistency" emerges due to local updates (Karimireddy et al., 2020; Wang et al., 2020), the full/uniform client participation in each communication round averages them out in the long run, therefore, guaranteeing convergence.

A related interesting line of works in FL different from full/uniform client participation focuses on *proactively creating* flexible client participation (see, e.g., (Xie et al., 2019; Ruan et al., 2021; Gu et al., 2021; Avdiukhin & Kasiviswanathan, 2021; Yang et al., 2022; Wang & Ji, 2022)). The main idea here is to allow asynchronous communication or fixed participation pattern (e.g., given probability) for clients to flexibly participate in training. Existing works in this area often require extra assumptions, such as bounded delay (Ruan et al., 2021; Gu et al., 2021; Yang et al., 2022; Avdiukhin & Kasiviswanathan, 2021) and identical computation rate (Avdiukhin & Kasiviswanathan, 2021). Under these assumptions, although the stochastic gradients are no longer unbiased estimators of the

full gradients, the deviation in each communication round remains bounded. For sufficiently many communication rounds, the impact of such deviation from full gradients vanishes asymptotically, since each client can *still* participate in FL in the long run. In contrast, this paper considers a more practical (or worst-case) scenario in FL – *partial client participation even in the long run*, which can be caused by many real-world system heterogeneity factors as mentioned in Section 1 .

**2) Domain Adaptation:** Since partial client participation induces a gap between the dataset distribution used for FL training and the true data population distribution across all clients, our work is also related to the field of domain adaptation in learning. Domain adaptation focuses on the learnability of a model trained in one source domain but applied to a different and related target domain. The basic approach is to quantify the error in terms of the source domain plus the distance between source and target domains. Specifically, let $P$ and $Q$ be the target and source distributions, respectively. Then, the generalization error is expressed as $\mathcal{O}(\mathcal{A}(n_Q)) + dist(P, Q)$, where $\mathcal{A}(n_Q)$ is an upper bound of the error dependent on the total number of samples in $Q$, Widely-used distance measures include $d_\mathcal{A}$-divergence (Ben-David et al., 2010; David et al., 2010) and $\mathcal{Y}$-discrepancy (Mansour et al., 2009; Mohri & Medina, 2012). We note, however, that results in domain adaptation cannot be directly applied in FL with partial client participation, since doing so yields an overly pessimistic bound. Specifically, the error based on domain adaptation remains non-zero for asymptotically small distance $dist(P, Q)$ between $P$ and $Q$ even with infinite many samples in $n_Q$ (i.e., $\mathcal{A}(n_Q) \to 0$). In this paper, rather than directly using results from domain adaptation, we establish a much *sharper* upper bound (see Section 3). Another line of work in domain adaptation is using importance weights defined by the density ratios between $P$ and $Q$ to correct the bias and reduce the discrepancy (Sugiyama et al., 2007a;b; Cortes et al., 2008). However, due to FL privacy constraints, such density ratios are difficult to estimate, which renders importance-weights-based methods infeasible in FL. A closely related work is (Hanneke & Kpotufe, 2019), which proposed a new notion of discrepancy between source and target distributions called transfer exponents. However, this work considers *non-overlapping* support between $P$ and $Q$, while we focus on *overlapping* support (see Fig. 1 in Section 3.3).

# 3 PAC-LEARNABILITY OF CONVENTIONAL FEDERATED LEARNING WITH PARTIAL CLIENT PARTICIPATION

In this section, we first focus on understanding the impacts of partial client participation on conventional FL in terms of PAC-learnability in Section 3.2. This will also pave the way for studying SA-FL later in Section 3.3. In what follows, we start with the conventional FL formulation and some definitions in statistical learning that are necessary to formulate and prove our main results.

## 3.1 PRELIMINARIES

The goal of FL is to minimize the following loss function $F(\mathbf{x}) = \mathbb{E}_{i \sim \mathcal{P}}[F_i(\mathbf{x})]$, where $F_i(\mathbf{x}) \triangleq \mathbb{E}_{\xi \sim P_i}[f_i(\mathbf{x}, \xi)]$. Here, $\mathcal{P}$ represents the distribution of the entire client population, $\mathbf{x} \in \mathbb{R}^d$ is the model parameter, $F_i(\mathbf{x})$ represents the local loss function at client $i$, and $P_i$ is the underlying distribution of the local dataset at client $i$. In general, $P_i \neq P_j$, if $i \neq j$ due to data heterogeneity. However, the loss function $F(\mathbf{x})$ or full gradient $\nabla F(\mathbf{x})$ can not be directly computed as the exact distribution of data is unknown in general. Instead, one often considers the following empirical risk minimization (ERM) problem in the form of finite-sum:

$$\min_{\mathbf{x} \in \mathbb{R}^d} \hat{F}(\mathbf{x}) = \sum_{i \in [M]} \alpha_i \hat{F}_i(\mathbf{x}),$$

where $\hat{F}_i(\mathbf{x}) \triangleq (1/|S_i|) \sum_{\xi \in S_i} f_i(\mathbf{x}, \xi)$. Here, $M$ is the total number of clients, $S_i$ is the local dataset with cardinality $|S_i|$, which is i.i.d. and sampled from distribution $P_i$, $\alpha_i = |S_i|/(\sum_{j \in [M]} |S_j|)$ (hence $\sum_{i \in [M]} \alpha_i = 1$). For ease of presentation, we consider the balanced dataset case: $\alpha_i = 1/M, \forall i \in [M]$. Next, we state several definitions from statistical learning theory (Mohri et al., 2018).

**Definition 1** (Generalization Error and Empirical Error). *Given a hypothesis $h \in \mathcal{H}$, a target concept $f$, an underlying distribution $\mathcal{D}$ and a dataset $S$ i.i.d. sampled from $\mathcal{D}$ ($S \sim \mathcal{D}$), the generalization error and empirical error of $h$ are defined as follows: $\mathcal{R}_\mathcal{D}(h, f) = \mathbb{E}_{(x,y) \sim \mathcal{D}} l(h(x), f(x))$ and $\hat{\mathcal{R}}_D(h, f) = \frac{1}{|S|} \sum_{i \in S} l(h(x_i), f(x_i))$, where $l(\cdot)$ is some valid loss function.*

With a slight abuse of notation, we omit $f$ by using $\mathcal{R}_\mathcal{D}(h)$ and $\hat{\mathcal{R}}_D(h)$ for simplicity.

**Definition 2** (Optimal Hypothesis). *For a distribution $\mathcal{D}$ and a dataset $S \sim \mathcal{D}$, we define $h_\mathcal{D}^* = \underset{h \in \mathcal{H}}{\text{argmin}} \mathcal{R}_\mathcal{D}(h)$ and $\hat{h}_\mathcal{D}^* = \underset{h \in \mathcal{H}}{\text{argmin}} \hat{\mathcal{R}}_\mathcal{D}(h)$.*

**Definition 3** (Excess Error). *For hypothesis $h$ and distribution $\mathcal{D}$, the excess error and excess empirical error are defined as $\varepsilon_\mathcal{D}(h) = \mathcal{R}_\mathcal{D}(h) - \mathcal{R}_\mathcal{D}(h_\mathcal{D}^*)$, and $\hat{\varepsilon}_\mathcal{D}(h) = \hat{\mathcal{R}}_\mathcal{D}(h) - \hat{\mathcal{R}}_\mathcal{D}(\hat{h}_\mathcal{D}^*)$, respectively.*

### 3.2 CONVENTIONAL FEDERATED LEARNING UNDER PARTIAL CLIENT PARTICIPATION

With the above notations, we consider conventional FL under partial client participation. Consider an FL system with $M$ clients in total. We let $P$ denote the underlying joint distribution of the entire system, which can be decomposed into the summation of the local distributions at each client, i.e., $P = \sum_{i \in [M]} \lambda_i P_i$, where $\lambda_i > 0$ and $\sum_{i \in [M]} \lambda_i = 1$. We assume that each client $i$ has $n$ training samples i.i.d. drawn from $P_i$, i.e., $|S_i| = n, \forall i \in [M]$. Then, $S = \{(x_i, y_i), i \in [M \times n]\}$ can be viewed as the dataset i.i.d. sampled from the joint distribution $P$. We consider a partial client participation setting, where $m \in [0, M)$ clients participate in the FL training as a result of some client sampling process $\mathcal{F}$. We let $\mathcal{F}(S)$ represent the data ensemble actually used in training and $\mathcal{D}$ denote the underlying distribution corresponding to $\mathcal{F}(S)$. For convenience, we define the notion $\alpha = \frac{m}{M}$ as the *FL system capacity* (i.e., only $m$ clients participate in the training). For FL with partial client participation, we establish the following fundamental performance limit of any learner in general. For simplicity, we use binary classification with zero-one loss here. We state the following impossibility result in Theorem 1 in terms of PAC learnability:

**Theorem 1** (Impossibility Theorem). *Let $\mathcal{H}$ be a non-trivial hypothesis space and $\mathcal{L}$ : $(\mathcal{X}, \mathcal{Y})^{(m \times n)} \to \mathcal{H}$ be the learner for an FL system. There exists a client participation process $\mathcal{F}$, a distribution $P$, and a target function $f \in \mathcal{H}$ with $\min_{h \in \mathcal{H}} \mathcal{R}_P(h, f) = 0$, such that $\mathbb{P}_{S \sim P}[\mathcal{R}_P(\mathcal{L}(\mathcal{F}(S), f)) > \frac{1 - \alpha}{8}] > \frac{1}{20}$.*

*Proof Sketch.* The proof is based on the method of induced distributions in (Bshouty et al., 2002; Mohri et al., 2018; Konstantinov et al., 2020). We first show that the learnability of an FL system is equivalent to that of a system that arbitrarily selects $mn$ out of $Mn$ samples in the centralized learning. Then, for any learning algorithm, there exists a distribution $P$ such that dataset $\mathcal{F}(S)$ resulting from partial participation and seen by the algorithm is always distributed identically for any target functions. Thus, no algorithm can learn a better predictor than random guessing. Due to space limitation, we relegate the full proof to supplementary material. □

Given the system capacity $\alpha \in (0, 1)$, the above theorem characterizes the worst-case scenario for FL with partial client participation. It says that for any learner (i.e., algorithm) $\mathcal{L}$, there exists a bad client participation process $\mathcal{F}$ and distributions $P_i, i \in [M]$ over target function $f$, for which the error of the hypotheses returned by $\mathcal{L}$ is constant with non-zero probability. In other words, FL with partial client participation is *not PAC-learnable*. One interesting observation here is that the lower bound is *independent* of the number of samples per client $n$. This indicates that even if each client has *infinitely many* samples ($n \to \infty$), it is impossible to have a zero-generation-error learner under the partial client participation ($\alpha \in (0, 1)$). Note that this fundamental result relies on two conditions: *heterogeneous* dataset and *arbitrary* client participation. Under these two conditions, there exists a worst-case scenario where the underlying distribution $\mathcal{D}$ of the participating data $S_\mathcal{D} = \mathcal{F}(S)$ deviates from the ground-truth $P$, thus yielding a non-vanishing error. This result also sheds light on how to motivate client participation in FL effectively and efficiently: the participating client's data should be comprehensive enough to model the complexity of the joint distribution $P$ to close the gap between $\mathcal{D}$ and $P$. Note that this result is not contradictory to previous works where the convergence of FedAvg is guaranteed, since this theorem is not applicable for homogeneous (i.i.d.) datasets or uniformly random client participation. As mentioned earlier, most of the existing works rely on at least one of these two assumptions. However, none of these two assumptions hold for conventional FL with partial client participation in practice.

### 3.3 THE PAC-LEARNABILITY OF SERVER-AIDED FEDERATED LEARNING (SA-FL)

The intuition of SA-FL is to utilize a dataset $T$ i.i.d. sampled from distribution $P$ with cardinality $n_T$ as a vehicle to correct potential distribution derivations due to partial client participation. By doing

so, the server steers the learning by a small number of representative data, while the clients aid the learning by federation to leverage the huge amount of privately decentralized data ($n_S \gg n_T$). Note that the assumption of access to this dataset is not restrictive since such datasets are already available in many FL systems: although not always necessary for training, an auxiliary dataset is often needed for defining FL tasks (e.g., simulation prototyping) before training and model checking after training (e.g., quality evaluation and sanity checking) (McMahan et al., 2021; Wang et al., 2021a). Also, obtaining an auxiliary dataset is affordable since the number of data points required is relatively small (of the order of hundreds, see our experimental results), and hence the cost is low. Then, SA-FL can be easily achieved or even with manually labelled data thanks to its small size. It is also worth noting that many works use such auxiliary datasets in FL for security (Cao et al., 2021), incentive design (Wang et al., 2019), and knowledge distillation (Cho et al., 2021).

For SA-FL, we consider the same partial client participation setting that induces a dataset $S_{\mathcal{D}} \sim \mathcal{D}$ with cardinality $n_S$ and $\mathcal{D} \neq P$. As a result, the learning process is to minimize $\mathcal{R}_P(h)$ by utilizing $(\mathcal{X}, \mathcal{Y})^{n_T + n_S}$ to learn a hypothesis $h \in \mathcal{H}$. For notional clarity, we assume the joint dataset $S_Q = (S_{\mathcal{D}} \cup T) \sim Q$ with cardinality $n_T + n_S$ for some distribution $Q$. Before deriving the generalization error bound for SA-FL, we state the following assumption and definition.

**Assumption 1** (Noise Condition). *Suppose that $h_P^*$ and $h_Q^*$ exist. There exist $\beta_P, \beta_Q \in [0,1]$ and $\alpha_P, \alpha_Q > 0$ such that the following hold: $\mathbb{P}_{x \sim P}(h(x) \neq h_P^*(x)) \leq \alpha_P[\varepsilon_P(h)]^{\beta_P}$, $\mathbb{P}_{x \sim Q}(h(x) \neq h_Q^*(x)) \leq \alpha_q[\varepsilon_Q(h)]^{\beta_Q}$.*

This assumption is a traditional noise model known as the Bernstein class condition, which has been widely used in the literature (Massart & Nédélec, 2006; Koltchinskii, 2006; Hanneke, 2016).

**Definition 4** (($\alpha, \beta$)-Positively-Related). *Distributions $P$ and $Q$ are said to be ($\alpha, \beta$)-positively-related if there exist constants $\alpha \geq 0$ and $\beta \geq 0$ such that $|\varepsilon_P(h) - \varepsilon_Q(h)| \leq \alpha[\varepsilon_Q(h)]^{\beta}, \forall h \in \mathcal{H}$.*

Definition 4 specifies a stronger constraint between distributions $P$ and $Q$. It indicates that the difference of excess error for one hypothesis $h \in \mathcal{H}$ between $P$ and $Q$ is bounded by the excess error of $Q$ in some exponential form. With the above assumption and definition, we have the following generation error bound for SA-FL, which shows that SA-FL is PAC-learnable:

**Theorem 2** (Generalization Error Bound for SA-FL). *For an SA-FL system with arbitrary system and data heterogeneity, if distributions $P$ and $Q$ satisfy Assumption 1 and are ($\alpha, \beta$)-positively-related, then with probability at least $1 - \delta$ for any $\delta \in (0,1)$, it holds that*

$$\varepsilon_P(\hat{h}_Q^*) = \widetilde{\mathcal{O}}\left( \left( \frac{d_{\mathcal{H}}}{n_T + n_S} \right)^{\frac{1}{2 - \beta_Q}} + \left( \frac{d_{\mathcal{H}}}{n_T + n_S} \right)^{\frac{\beta}{2 - \beta_Q}} \right), \tag{1}$$

*where $d_{\mathcal{H}}$ denotes the finite VC dimension for hypotheses class $\mathcal{H}$, and parameters $\{P, Q, n_T, n_S, \beta, \beta_Q\}$ are defined the same as before.*

It is known that (see, e.g., (Hanneke, 2016)) the generalization error bound of centralized learning is $\widetilde{\mathcal{O}}((\frac{1}{n})^{\frac{1}{2 - \beta_Q}})$ (hiding logarithmic factors) with $n$ samples in total and noise parameter $\beta_Q$. Note that when $\beta \geq 1$, the first term in Eq. (1) dominates. Hence, Theorem 2 implies that the generalization error bound for SA-FL *matches* that of centralized learning (with dataset size $n_T + n_S$). Meanwhile, compared with solely training on server's dataset $T$, SA-FL exhibits an improvement from $\widetilde{\mathcal{O}}((\frac{1}{n_T})^{\frac{1}{2 - \beta_Q}})$ to $\widetilde{\mathcal{O}}((\frac{1}{n_T + n_S})^{\frac{\beta}{2 - \beta_Q}})$.

Note that SA-FL shares some similarity with the domain adaptation problem, where the learning is on $Q$ but the results will be adapted to $P$. In what follows, we offer some deeper insights between the two by answering two key questions: *1) What is the difference between SA-FL and domain adaptation (a.k.a. transfer learning)?* and *2) Why is SA-FL from $Q$ to $P$ PAC-learnable, but FL from $\mathcal{D}$ to $P$ with partial client participation not PAC-learnable (as indicated in Theorem 1)?*

To answer these questions, we illustrate the distribution relationships for domain adaptation and federated learning, in Fig. 1, respectively. In domain adaptation, the target $P$ and source $Q$ distributions often have overlapping support but there also exists *distinguishable difference*. In contrast, the two distributions $P$ and $Q$ in SA-FL happen to share exactly the *same support* with different density, since $Q$ is a *mixture* of $\mathcal{D}$ and $P$. As a result, the known bounds in domain adaptation (or transfer learning)

are pessimistic in SA-FL. For example, the $dist(P, Q)$ in $d_{\mathcal{A}}$-divergence and $\mathcal{Y}$-divergence both have non-negligible gaps when applied to SA-FL. Here in Theorem 2, we provide a generalization error bound in terms of the total sample size $n_T + n_S$, thus showing the benefit of SA-FL.

Moreover, for SA-FL, only the auxiliary dataset $T \overset{i.i.d.}{\sim} P$ is directly available for the server. The clients' datasets could be used in SA-FL training, but they are not directly accessible due to privacy constraints. Thus, previous methods in domain adaptation (e.g., importance weights-based methods in covariate shift adaptation (Sugiyama et al., 2007a;b)) are *not* applicable since they require the knowledge of density ratio between training and test datasets.

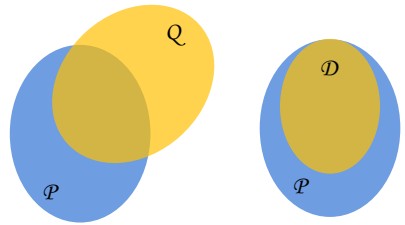

**Domain Adaptation    Federated Learning**

Figure 1: Diagram of distributions for domain adaptation and federated learning.

The key difference between FL and SA-FL lies in relations among $D$, $P$ and $Q$. For FL, the distance between $D$ and $P$ under partial participation could be large due to system and data heterogeneity in the worst-case. More specifically, the support of $D$ could be narrow enough to miss some part of $P$, resulting in non-vanishing error as indicated in Theorem 1. For SA-FL, distribution $Q$ is a mixture of $P$ and $D$ ($Q = \lambda_1 D + \lambda_2 P$, with $\lambda_1, \lambda_2 \geq 0$, $\lambda_1 + \lambda_2 = 1$), thus having the same support with $P$. Hence, under Assumption 4, the PAC-learnability is guaranteed.

Although we provide a promising bound to show the PAC-learnability of SA-FL in Theorem 2, the superiority of SA-FL over training solely with dataset $T$ in server (i.e., $\widetilde{\mathcal{O}}((\frac{1}{n_T})^{\frac{1}{2-\beta_P}})$) is not always guaranteed as $\beta \to 0$ (i.e., $Q$ becomes increasingly different from $P$). In what follows, we reveal under what conditions between $P$ and $Q$ could SA-FL perform *no worse than* centralized learning in terms of generalization error.

**Theorem 3** (SA-FL Being No Worse Than Centralized Learning). *Consider an SA-FL system with arbitrary system and data heterogeneity. If Assumption 1 holds and additionally $\hat{\mathcal{R}}_P(\hat{h}_Q^*) \leq \hat{\mathcal{R}}_P(h_Q^*)$ and $\varepsilon_P(h_Q^*) = \mathcal{O}(\mathcal{A}(n_T, \delta))$, where $\mathcal{A}(n_T, \delta) = \frac{d_{\mathcal{H}}}{n_T} \log(\frac{n_T}{d_{\mathcal{H}}} + \frac{1}{n_T} \log(\frac{1}{\delta}))$, then with probability at least $1 - \delta$ for any $\delta \in (0, 1)$, it holds that $\varepsilon_P(\hat{h}_Q^*) = \widetilde{\mathcal{O}}\left((d_{\mathcal{H}}/n_T)^{\frac{1}{2-\beta_P}}\right)$. Other parameters are the same as defined in Theorem 2.*

Here, we remark that $\varepsilon_P(h_Q^*) = \mathcal{O}(\mathcal{A}(n_T, \delta))$ is a weaker condition than the $\varepsilon_P(h_Q^*) = 0$ condition and the covariate shift assumption ($P_{Y|X} = Q_{Y|X}$) used in the transfer learning literatures (Hanneke & Kpotufe, 2019; 2020). Together with the condition $\hat{\mathcal{R}}_P(\hat{h}_Q^*) \leq \hat{\mathcal{R}}_P(h_Q^*)$, the following intermediate result holds: $\hat{\mathcal{R}}_P(\hat{h}_Q^*) - \hat{\mathcal{R}}_P(h_P^*) = \mathcal{O}(A(n_T, \delta))$ (see Lemma 2 in the supplementary material). Intuitively, this states that "if $P$ and $Q$ share enough similarity, then the difference of excess empirical error between $\hat{h}_Q^*$ and $h_P^*$ on $P$ can be bounded." Thus, the excess error of $\hat{h}_Q^*$ shares the same upper bound as that of $\hat{h}_P^*$ in centralized learning. Therefore, Theorem 3 implies that, under mild conditions, SA-FL guarantees the same generalization error upper bound as that of centralized learning with dataset $T$, hence being "no worse than" centralized learning with dataset $T$.

## 4    THE SAFARI ALGORITHM

Although we have shown that SA-FL is PAC-learnable under partial client participation, we note that SA-FL still follows the server-client architecture, which demands intensive communication between the server and clients. Hence, it is important to design communication-efficient algorithms for SA-FL with a comparable level of communication complexity as that of conventional FL. In this section, we propose a new algorithm called SAFARI (server-aided federated averaging) algorithm for SA-FL and characterize its convergence guarantees.

As shown in Algorithm 1, SAFARI iteratively performs the following three steps: 1) Server samples a subset of clients as in conventional FL and synchronize the latest global model $\mathbf{x}_r$ with each participating clients (Line 3); 2) The server and all participating clients train the model based on local dataset (Lines 4-8). Specifically, client initializes its local model with $\mathbf{x}_r$ and then performs $K$ local steps by the stochastic gradient descent method. Then, each client sends its locally accumulated

---

**Algorithm 1** The SAFARI Algorithm for SA-FL.

---

1: Initialize $\mathbf{x}_0$.
2: **for** $r = 0, \cdots, R - 1$ **do**
3:     The server samples a subset $S_r$ of clients with $|S_r| = n$ and send current model $x_t$.
4:     **for** Each client $i \in S_r$ **do**
5:         Synchronization: $\mathbf{x}_{r,0}^i = \mathbf{x}_r$.
6:         Local updates: for $k = 0, ..., K - 1$:     $\mathbf{x}_{r,k+1}^i = \mathbf{x}_{r,k}^i - \eta \nabla F_i(\mathbf{x}_{r,k}^i, \xi_{r,k}^i)$.
7:         Send $\Delta_r^i = -\sum_{k=0}^{K-1} \nabla F_i(\mathbf{x}_{r,k}^i, \xi_{r,k}^i)$ to server.
8:     **end for**
9:     **for** Server **do**
10:         Local updates: for $k = 0, ..., K - 1$:     $\mathbf{x}_{r,k+1}^0 = \mathbf{x}_{r,k}^0 - \eta \nabla F(\mathbf{x}_{r,k}^0, \xi_{r,k}^0)$,
           $\Delta_r^0 = -\sum_{k=0}^{K-1} \nabla F(\mathbf{x}_{r,k}^0, \xi_{r,k}^0)$.
11:         Receive $\Delta_r^i, i \in S_r$, and normalize it:     $\hat{\Delta}_r^i = c_r \frac{\Delta_r^i - \Delta_r^0}{\|\Delta_r^i - \Delta_r^0\|}$.
12:         Server Update:
           $\mathbf{x}_{r+1} = \mathbf{x}_r + \eta \left( \Delta_r^0 + \frac{1}{|S_r|} \sum_{i \in S_r} \hat{\Delta}_r^i \right)$.
13:     **end for**
14: **end for**

---

update $\Delta_r^i$ back to the server. Note the server simultaneously takes $K$ local steps based on its auxiliary dataset (in Line 10). 3) Server aggregates and updates the global model (Lines 11-12). Upon receiving the local update $\Delta_r^i$, the server normalizes and rescales it by a hyper-parameter $c_r$. Then, the server updates the global model by aggregating the normalized update $\hat{\Delta}_r^i$ and the server's update $\Delta_r^0$ based on its own auxiliary dataset. Compared to FedAvg (McMahan et al., 2017) in FL, SAFARI shares the same communication and computation process from the client's perspective. Hence, it enjoys the same level of communication efficiency and privacy benefits. Before conducting the convergence performance analysis, we first state two commonly used assumptions in the FL literature.

**Assumption 2.** *(L-Lipschitz Continuous Gradient) There exists a constant $L > 0$, such that* $\|\nabla F(\mathbf{x}) - \nabla F(\mathbf{y})\| \leq L\|\mathbf{x} - \mathbf{y}\|, \forall \mathbf{x}, \mathbf{y} \in \mathbb{R}^d$.

**Assumption 3.** *(Unbiased Stochastic Gradient with Bounded Variance) The stochastic gradient is unbiased, i.e.,* $\mathbb{E}[\nabla F(\mathbf{x}, \xi)] = \nabla F(\mathbf{x})$ *and* $\mathbb{E}[\|\nabla F(\mathbf{x}, \xi) - \nabla F(\mathbf{x})\|^2] \leq \sigma^2$.*partial*

With the assumptions above, we are now in the position to analyze the convergence of SAFARI.

**Theorem 4** (Convergence Rate for SAFARI ). *Under Assumptions 2 and 3, let constant learning rate $\eta$ satisfy $(\frac{1}{2} - 4LK\eta - 20K(L + 4KL^3\eta)\eta^2) > 0$. Then, the sequence $\{\mathbf{x}_r\}$ generated by the* SAFARI *algorithm satisfies:*

$$\frac{1}{R} \sum_{t=0}^{R-1} \mathbb{E}\|\nabla F(\mathbf{x}_r)\|^2 \leq \frac{1}{c} \left[ \frac{F(\mathbf{x}_0) - F(\mathbf{x}^*)}{\eta K R} \right] + \frac{1}{c} \left[ \left( 5KL\eta^2 + 20K^2L^3\eta^3 + 2L\eta \right) \sigma^2 \right]$$

$$+ \frac{1}{c} \left[ \left( \frac{1}{K^2} + \frac{L\eta}{K} \right) \frac{1}{R} \sum_{t=0}^{R-1} c_r^2 \right],$$

*where $c$ is a constant and $\mathbf{x}^*$ denotes an optimal solution.*

Theorem 4 implies an $\mathcal{O}(1/R)$ convergence rate to a neighborhood of a stationary point. Furthermore, by choosing parameters $\{c_r\}$ and the learning rate $\eta$ appropriately, we have the following convergence rate to a stationary point:

**Corollary 1.** *If $\sum_{t=0}^{R-1} c_r^2$ is bounded and learning rate $\eta = \frac{1}{\sqrt{KR}}$, the convergence rate of* SAFARI *is:*

$$\mathcal{O} \left( \frac{1}{K^{1/2}R^{1/2}} + \frac{1}{R} + \frac{1}{K^2 R} + \frac{1}{K^{3/2}R^{3/2}} + \frac{K^{1/2}}{R^{3/2}} \right).$$

In Theorem 4 and Corollary 1, we show the convergence guarantee of SAFARI under no extra assumptions on the data and system heterogeneity (client participation), which corroborates the learnability

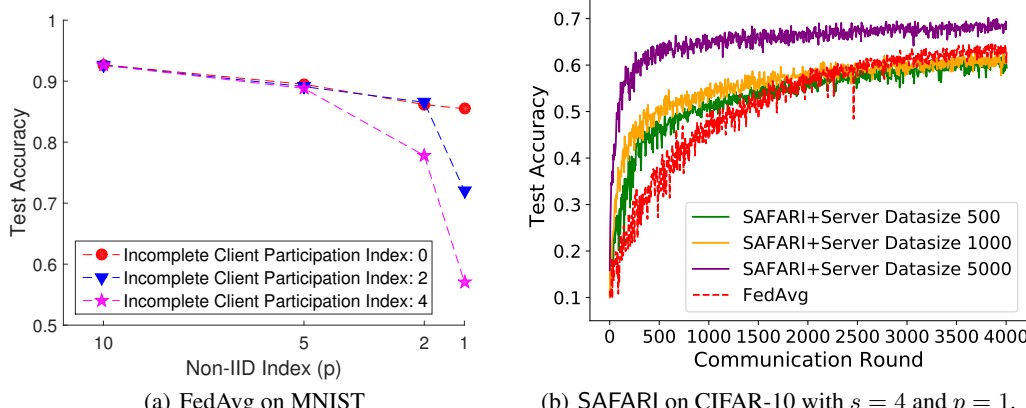

(a) FedAvg on MNIST      (b) SAFARI on CIFAR-10 with $s = 4$ and $p = 1$.

Figure 2: Test accuracy of FedAvg on MNIST and SAFARI on CIFAR-10 with partial client participation. Larger partial client participation index $s$ means less clients participate in the training, and smaller non-i.i.d. index $p$ means the data across clients are more heterogeneous.

analysis in Section 3. Hence, in the worst-case scenarios, convergence rate $\mathcal{O}(1/(K^{1/2}R^{1/2}))$ is achieved for sufficiently large $R$ and $K \leq R$. In comparison, a non-vanishing error term emerges consistently for the same setting in FL (Yang et al., 2022). This verifies the superiority of SA-FL over conventional FL with partial client participation. Note that Corollary 1 requires a convergent series $\{c_r^2\}$. This can be relaxed to $\sum_{r=0}^{R-1} c_r^2 = \mathcal{O}(\min\{KR, K^{3/2}R^{1/2}\})$ to maintain the same $\mathcal{O}(1/(K^{1/2}R^{1/2}))$ rate. It can be readily verified that $p$-series ($c_r = r^{-p}$) satisfies the condition.

Also, we will see later in numerical results that, compared to FedAvg, our SAFARI has a faster convergence as the size of auxiliary server dataset increases. It is also worth noting that SA-FL can be used to mitigate Byzantine attacks to FL systems. For example, empirical success of utilizing an auxiliary dataset and normalization at the server has been shown in (Cao et al., 2021) in defending against Byzantine attacks in FL systems. Lastly, note that the convergence rate bound in Corollary 1 does not explicitly depend on the number of participating clients $m$. It is thus an interesting future work to study whether there exist algorithms for SA-FL that theoretically guarantee a "linear speedup effect" as the number of participating clients increases under partial client participation.

## 5 NUMERICAL RESULTS

In this section, we conduct numerical experiments to verify our theoretical results using 1) logistic regression (LR) on MNIST dataset (LeCun et al., 1998) and 2) convolutional neural network (CNN) on CIFAR-10 dataset (Krizhevsky et al., 2009). To simulate data heterogeneity, we distribute the data into each client evenly in a label-based partition, following the same process as in previous works (e.g., (McMahan et al., 2017; Yang et al., 2021b; Li et al., 2020b)). As a result, we can use a parameter $p$ to represent the classes of labels in each client's dataset, which serves as an index of data heterogeneity level (non-i.i.d. index). The smaller $p$-value, the more heterogeneous the data among clients. To mimic partial client participation, we force $s$ clients to be excluded. We can use $s$ as an index to represent the degree of such partial client participation. In our experiments, there are $M = 10$ clients in total, and $m = 5$ clients participate in the training in each communication round, who are uniformly sampled from the $M - s$ clients. We use two algorithms as baselines to compare with our algorithm: i) the FedAvg algorithm without any auxiliary dataset at the server; and ii) the SGD algorithm for centralized learning with dataset only at the server (without any clients). Due to space limitation, we highlight four key observations in this section, and relegate the details of the experiments and additional experiments to the supplementary material.

**1) Performance Degradation of Partial Client Participation:** We first show the test accuracy of FedAvg on MNIST for different values of non-i.i.d. index $p$ and partial client index $s$ in Fig. 2(a). For nearly homogeneous data (e.g., from $p = 10$ to $p = 5$), partial client participation has negligible impacts on test accuracy. However, for highly non-i.i.d. cases, partial client participation results in dramatic performance degradation. Specifically, for $p = 1$, the test accuracy for $s = 4$ is only 57%,

Table 1: Test accuracy improvement (%) for SAFARI compared with FedAvg on MNIST with partial client participation $s = 4$. '-' means "no statistical difference within 2% error bar".

| SERVER | NON-IID INDEX ($p$) | | | |
|--------|----|----|------|------|
| DATASIZE | 10 | 5 | 2 | 1 |
| 50 | - | - | - | 12.32 |
| 100 | - | - | 5.24 | 16.48 |
| 500 | - | - | 9.40 | 27.55 |
| 1000 | - | - | 10.08 | 28.78 |

Table 2: Test accuracy improvement (%) for SAFARI under partial client participation $s = 4$ compared with SGD in centralized learning on MNIST. Smaller non-i.i.d. index means the data across clients is more heterogeneous. '-' means "no statistical difference within 2% error bar".

| SERVER | NON-IID INDEX ($p$) | | | |
|--------|-------|-------|------|------|
| DATASIZE | 10 | 5 | 2 | 1 |
| 50 | 28.94 | 19.65 | 9.32 | 3.69 |
| 100 | 20.46 | 14.84 | 8.58 | -3.42 |
| 500 | 6.13 | 3.92 | 2.09 | - |
| 1000 | 3.79 | 3.16 | - | - |

yielding a large degradation (35%) compared to that of $s = 0$. This is consistent with the worst-case analysis in Theorem 1 and also the main motivation of SA-FL.

**2) Improvement of the** SAFARI **Algorithm under Partial Client Participation:** In Table 1, we show the test accuracy improvement of our SAFARI algorithm compared with that of FedAvg in standard FL. The key observation is that, with a *small amount* of auxiliary data at the server, there is a significant increase of test accuracy for our SAFARI algorithm. For example, with only 50 data samples at the server (0.1% of the total training data), there is a 12.32% test accuracy increase. With 500 data samples, the improvement reaches 27.55%. This verifies the effectiveness of our SA-FL framework and our SAFARI algorithm. Another observation is that for nearly homogeneous case (e.g., from $p = 10$ to $p = 5$), there is no statistical difference with or without auxiliary data at the server (denoted by '-' in Table 1. This is consistent with the previous observations of negligible degradation in cases with homogeneous data across clients.

**3) Benefits of Collaboration Even with Partial Client Participation:** In Table 2, we show the comparison between our SAFARI algorithm under partial client participation ($s = 4$) and SGD for centralized learning (i.e., only with data at the server). This comparison is to illustrate the benefit of collaborations from the clients. We can see these collaborations from the clients significantly improve the performance for the nearly homogeneous case, especially when the size of the server dataset is small (cf. the columns of $p = 10$ and $p = 5$). However, for highly heterogeneous cases, it shows no obvious improvement from the collaboration of clients if participation is partial. This confirms our theoretical analysis in Theorem 2. Specifically, highly partial client participation and heterogeneous data render a small $\beta$-value, and so there is no clear benefit of collaboration in FL over centralized learning only on small auxiliary dataset due to the significant distribution deviation.

**4) Convergence Speedup of** SAFARI **with Larger Server Dataset:** In this experiment, we illustrate the speedup effect of SAFARI numerically as the size of server dataset increases. In Fig. 2(b), we show the convergence processes of SAFARI on CIFAR-10 for partial client participation ($s = 4$) and non-i.i.d. data ($p = 1$). We can see clearly that the convergence of SAFARI is accelerating and the test accuracy increases as more data are employed at the server. In this experiment setting, we also plot the convergence of FedAvg in Fig. 2(b). It can be seen that all three cases of SAFARI converge faster than FedAvg in this experiment.

## 6 CONCLUSION

In this paper, we rigorously investigated the server-aided federated learning (SA-FL) framework, which has been increasingly adopted in practice to mitigate the impacts of partial client participation in conventional FL. In SA-FL, the key idea is to deploy an auxiliary dataset at the server to reduce the distribution deviation induced by partial client participation. To characterize the benefits of SA-FL, we first showed that conventional FL is *not* PAC-learnable under partial client participation by establishing a fundamental generalization error lower bound. Then, we showed that SA-FL is able to revive the PAC-learnability of conventional FL under partial client participation. Upon resolving the PAC-learnability challenge, we proposed a new SAFARI (server-aided federated averaging) algorithm that enjoys convergence guarantee and the same level of communication efficiency as that of conventional FL. Extensive numerical results also validated our theoretical findings.

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
