# OpenReview forum: "On the Efficacy of Server-Aided Federated Learning against Partial Client Participation"
_ICLR.cc/2023/Conference — Submitted to ICLR 2023_

### Official Review · Reviewer_dvBg · 2022-11-02

**Confidence:** 4
**Clarity, Quality, Novelty And Reproducibility:** .
**Correctness:** 1
**Technical Novelty And Significance:** 3
**Empirical Novelty And Significance:** 3
**Recommendation:** 3

**Strength And Weaknesses:**

Before I start, I should acknowledge that I'm not an expert in statistical learning theory, so in this part of the paper, I can easily miss some important things. I am closer to the theory of optimization methods.

The first part is devoted to the statistical learning theory of FL under the partial participation setting. The motivation is clear and very important. In the beginning, the authors prove the "Impossibility Theorem" which is very interesting. It states the vanilla FedAvg will fail under the authors' setting. So that is why the authors analyze the idea of SA-FL.

From this point, I see the first weakness that I would like the authors to clarify:

1. The server has access to the dataset $T.$ So, in total, we have access to the dataset $Q = D \cup T.$ Not going into the details, the motivation of SA-FL is based on two assumptions: Assumption 1 and Definition 4, which introduce some similarity between datasets $Q$ and $P.$ The main motivation of the paper is to consider the case when $D \neq P.$ But from the assumption intuitively the authors require that $D \approx P.$ Moreover, is it possible to provide the "Impossibility Theorem" under these assumptions? The assumptions can be too strong.
2. Theorem 3: the assumption $\varepsilon_P(h^*_Q) \leq O(A(...))$ seems to be strong. They say that the optimal hypothesis from $Q$ is almost the optimal hypothesis from $P.$

In total, the statistical part I would evaluate with "marginally above the acceptance threshold."

Now, I move to Section 4.
The authors provide the SAFARI algorithm that has the following weaknesses:
1. From Theorem 4, it is clear that we have to take $c_t = 0$ to get the best convergence rate. It is not clear why we should take $c_t > 0.$
2. In line 3, the algorithm samples nodes uniformly. The main idea of the paper is to consider the setting with more complex samplings such that some nodes do not participate!
3. In line 10, the algorithm calculates gradients of the full function $F!$ The main point of FL is that server does not have access to $F!$ In the proof, the authors use the fact that $E_T \nabla F(x, \xi^0) = \nabla F(x) = E_P \nabla F(x, \xi).$ But in the previous section, the authors say that the server has access only to a small subset $T = (\xi^0_i)_i^k,$ so $E_T \nabla F(x, \xi^0) = \frac{1}{k} \sum_i \nabla F(x, \xi^0_i)$ and, in general, it does not equal to  $\nabla F(x).$ All it means is that the server has access to the function $F.$

Theoretically, the SAFARI algorithm does not solve the main problem of the paper.


**Summary Of The Paper:**

The authors consider FL problem in the scenario where some nodes do not participate in the learning and provide a theory that analysis this scenario. Moreover, the authors consider SA-FL: the server takes part in the optimization process to mitigate the corruption from partial participation of the nodes.

**Summary Of The Review:**

The statistical part (Section 3) of the paper seems to be solid. However, I have some questions about Assumption 1 and Definition 4 that are not fully justified.

At the same time, the optimization part (Section 4) is weak and does not provide a method that would solve the main problem.

Section 3 seems to be somewhere between "marginally above the acceptance threshold" and "accept, good paper." But Section 4 definitely should be improved.

---

> ### Author Response · Authors · 2022-11-16
> **Response to Reviewer dvBg [1/2]**
>
> Thank you very much for your review and constructive comments, which helped us significantly improve the quality of this paper. In this revision, we have carefully revised our paper based on your comments, questions, and suggestions. The detailed point-by-point responses are as follows:
>
> > **Your Comment 1:** The server has access to the dataset $T$. So, in total, we have access to the dataset $Q = D \cup T$. Not going into the details, the motivation of SA-FL is based on two assumptions: Assumption 1 and Definition 4, which introduce some similarity between datasets $Q$ and $P$. The main motivation of the paper is to consider the case when $D \neq P$. But from the assumption intuitively the authors require that $D \approx P$. Moreover, is it possible to provide the "Impossibility Theorem" under these assumptions? The assumptions can be too strong.
>
> **Our Response:** Thanks for your comments. It appears there is some confusion here and we would like to further clarify. First, the reviewer is correct that our main interest in this paper is to consider the case $D \ne P$. Also, the review's understanding is correct that, with $Q = D \cup T$ under SA-FL and $Q \approx P$ (due to Assumption 1 and Definition 4), we can show that SA-FL is PAC-learnable. This result provides a theoretical justfication to the use of SA-FL, which has been empirically observed to be effective to mitigate partial client participation in FL. However, it is *incorrect* to say that Assumption 1 and Definition 4 imply $D \approx P$. In fact, $D$ could be quite different from $P$ (see our illustration in Figure 1 in the paper). In essence, for SA-FL to be PAC-learnable, we *only* require $Q \approx P$. Thus, how good the server-side dataset $T$ should be to allow $Q = D \cup T \approx P$ is really critical. Assumption 1 and Definition 4 provide such theroetical sufficient conditions to characterize *"how good $T$ should be."*
>
> Regarding your "Impossibility Theorem" question, we guess you meant whether or not we could prove "Impossibility Theorem" if we replace "$Q$" by "$D$" in Assumption 1 and Definition 4, right? If our understanding of your question is correct, then the answer is "yes." This is because if $D \approx P$ in the sense of Assumption 1 and Definition 4, then the same PAC-learnable result can also be established for conventional FL. Thus, there will be *no* "Impossibility Theorem." However, as pointed out above, in SA-FL, $Q \approx P$ does **not** imply $D \approx P$.
>
> Lastly, we want to point out that the noise condition in Assumption 1, known as Bernstein noise condition, is a standard and widely used assumption in statistical learning with noise, as stated in the paper. The "$(\alpha,\beta)$-Positively-Related" condition in Definition 4 is directly motivated by and can be viewed as a generalization of the noise condition in Assumption 1. Specifically, motivated by the mathematical construct in Assumption 1, we propose to characterize the gap between the excess errors of *two* distributions $P$ and $Q$. Please see our responses to Reviewers uhzS and m3ZD, which explain why Assumption 1 and Definiton 4 are not too strong, respectively.
>
> > **Your Comment 2:** Theorem 3: the assumption $\epsilon_P(h^{*}_{Q}) \leq O(A(\dots))$ seems to be strong. They say that the optimal hypothesis from $Q$ is almost the optimal hypothesis from $P$.
>
> **Our Response:** Thanks for your comments. We note that the assumption $\epsilon_P(h^*_Q) = O(A(\dots))$ does *not* imply that "the optimal hypothesis from $Q$ is almost the optimal hypothesis from $P$." Here, we would like to further elaborate on how to interpret Theorem 3: Intuitively, the assumption $\epsilon_P(h^*_Q) = O(A(\dots))$ says that distributions $P$ and $Q$ should share some similarity, which is characterized by the expression $A(\dots)$. However, note that the $A(\dots)$-gap is just a Big-O result, which could be quite large, particularly when $n_T$ and $\delta$ are both small. This is completely different from the setting where "the optimal hypothesis from $Q$ is almost the optimal hypothesis of $P$," i.e., $\epsilon_P(h^{*}_Q) \approx 0$.
>
> [a]. Hanneke, Steve, and Samory Kpotufe. "On the value of target data in transfer learning." Advances in Neural Information Processing Systems 32 (2019).

---

> > ### Author Response · Authors · 2022-11-16
> > **Response to Reviewer dvBg [2/2]**
> >
> > > **Your Comment 3:** From Theorem 4, it is clear that we have to take $c_t = 0$ to get the best convergence rate. It is not clear why we should take $c_t > 0$.
> >
> > **Our Response:** Thanks for your comments. Note that setting $c_t=0$ corresponds to centralized learning *only* with server's dataset (see Algorithm 1, Line 11). Although it is true from Theorem 4 that the convergence rate is faster with $c_t=0$, the generalization performance of centralized training with a small server-side dataset is clearly worse than training with together with client-side data. In contrary, setting $c_t > 0$ allows us to utilize a large number of data from the clients, which leads to a better generalization performance.
> >
> >
> > > **Your Comment 4:** In line 3, the algorithm samples nodes uniformly. The main idea of the paper is to consider the setting with more complex samplings such that some nodes do not participate!
> >
> > **Our Response:** Thanks for your comments. This seems to be another misunderstanding, and we are happy to to further clarify. In Line 3 of Algorithm 1, the server only needs to sample a subset $S_t$ of clients with cardinaltiy $|S_t| = n$. Line 3 does *not* say that the sampling needs to be "uniform" (please check Line 3 in Algorithm 1). Here, the server does not require any sampling strategy, which could actually induce partial client participation. Also, our theoretical convergence analysis of Algorithm 1 is applicable for *any* sampling strategy including the case that some clients do not participate.
> >
> > > **Your Comment 5:** In line 10, the algorithm calculates gradients of the full function $F$! The main point of FL is that server does not have access to $F$! In the proof, the authors use the fact that $\mathbb{E}_T \nabla F(x, \xi^0) = \nabla F(x) = \mathbb{E}_P \nabla F(x, \xi)$. But in the previous section, the authors say that the server has access only to a small subset $T = (\xi_i^0)$ so $\mathbb{E}_T \nabla F(x, \xi^0) = \frac{1}{k} \sum_i \nabla F(x, \xi_i^0)$ and, in general, it does not equal to $\nabla F(x)$. All it means is that the server has access to the function $F$.
> >
> > **Our Response:** Thanks for your comments. There appears to be a misunderstanding here. In Line 10, the notation $\nabla F(x, \xi^0)$ represents a *stochastic gradient estimation* of function $F$ rather than the full gradient of $F$. Also, in our proof, we didn't mean $\mathbb{E}_T \nabla F(x, \xi^0) = \nabla F(x) = \mathbb{E}_P \nabla F(x, \xi)$ as you suggested. In our SAFARI algorithm, we construct $T$ by uniformly and independently sampling from $P$. This implies that the full expectation (over both sampling and the distribution of the obtained $T$) $\mathbb{E} [\nabla F(x, \xi^0)]$ is equal to the expectation of the sample mean of all samples in $T$, i.e., $\mathbb{E}[\frac{1}{k} \sum_i \nabla F(x, \xi_i^0)]$. This further implies that $\mathbb{E} [\nabla F(x, \xi^0)] = \mathbb{E}[\frac{1}{k} \sum_i \nabla F(x, \xi_i^0)] = \mathbb{E}[ \nabla F(x, \xi)] = \nabla F(x)$. Thus, the server only uses a stochastic estimation $\nabla F(x, \xi^0)$, which turns out to be an unbiased estimation of the full gradient $\nabla F(x)$ rather than the full gradient itself.

---

> > > ### Comment · Reviewer_dvBg · 2022-11-17
> > > **Respond**
> > >
> > > See my comments in the "Fatal Mistake" comment.

---

> ### Comment · Reviewer_dvBg · 2022-11-17
> **Fatal Mistake**
>
> Thank you! Due to the lack of time, only 1-2 days of the discussion are left, so I will blindly skip the discussion of "Comments" 1-2. In the future, please try to submit responses earlier.
>
> Let me move to "Comments" 3-5. Right now, Comments 3 and 4 are not so important because the author's understanding of Comment 5 contains a ***fatal mistake***. I will repeat the discussion here:
>
> I wrote:
>
> > In line 10, the algorithm calculates gradients of the full function $F!$ The main point of FL is that server does not have access to $F!$ In the proof, the authors use the fact that $E_T \nabla F(x, \xi^0) = \nabla F(x) = E_P \nabla F(x, \xi).$ But in the previous section, the authors say that the server has access only to a small subset $T = (\xi^0_i)_i^k,$ so $E_T \nabla F(x, \xi^0) = \frac{1}{k} \sum_i \nabla F(x, \xi^0_i)$ and, in general, it does not equal to  $\nabla F(x).$ All it means is that the server has access to the function $F.$
>
> The authors responded:
>
> > Thanks for your comments. Thanks for your comments. There appears to be a misunderstanding here. In Line 10, the notation  $\nabla F(x, \xi^0)$ represents a stochastic gradient estimation of function $F$ rather than the full gradient of $F$. ... In our SAFARI algorithm, we construct $T$ by uniformly and independently sampling from $P$...
> (see full comment in Response to Reviewer dvBg [2/2] )
>
> Let me explain where the authors are wrong. Let me consider the following procedure:
> 1. We take a dataset $P$
> 2. We define a function $F(x) = E_P F(x;\xi)$
> 3. We sample another dataset $T$ from $P$
> 4. We define a function $\widehat{F}(x) = E_T F(x;\xi^0)$
>
> I agree with you. For me it is clear that $E \widehat{F}(x) = F(x)$ and $E \nabla \widehat{F}(x) = \nabla F(x).$ But this is only true if $x$ is not a random vector!
>
> Let me consider the following SGD method:
>
> line 1: for $k = 1, ..., T$
>
> line 2: Sample $\xi^0$ from T
>
> line 3: $x^{k+1} = x^k - \gamma \nabla F(x^k;\xi^0)$
>
> We have $E_T \nabla F(x^k;\xi^0) = \nabla \widehat{F}(x^k),$ but $E \nabla F(x^k;\xi^0) = E \nabla \widehat{F}(x^k) \neq \nabla F(x^k).$ Note that $\nabla \widehat{F}(x^k) = \frac{1}{k} \sum_i \nabla F(x^k, \xi^0_i),$ where $\xi^0_i$ are random samples from $P$. You sample $\xi^0_i$ only ***once in the beginning*** of algorithms. It means that $x^k$ also depends on $\xi^0_i$, so $x^k$ is also random (even under the conditional expectation) and $E \nabla \widehat{F}(x^k) \neq \nabla F(x^k)$! The same problem is in SAFARI.
>
> I'm sorry, but I have to decrease my score because ***the paper contains a fatal mistake***. Till the end of the rebuttal, I'll be open to any discussion. And if the authors fix the mistake or convince me that I am not right, then I will increase the score back.

---

> > ### Author Response · Authors · 2022-11-18
> > **Response**
> >
> > **Our Response:** Thanks for your comments. There is actually no issue in our proof, and we belive that the confusion is probably caused by a misunderstanding of the expectation notation and the notation $T$.
> >
> > First and foremost, we want to clarify a confusion in our writing. In our submission, the notation $\nabla F(x^k)$ actually means the full expected value of the stochastic gradient $\nabla F(x^k)$ (note that the iterate $x^k$ is indeed random), which is averaged over i) all random trajectories $x^i$-iterates up to iteration $k$; and ii) the population distribution $P$. Thus, in our following response, we would like to explicitly rewrite this quantity as $\mathbb{E}[\nabla F(x^k)]$. We apologize for our earlier sloppy writing.
> >
> > Also, in our earlier discussions with the reviewer, we found that there is a slight abuse of notation $T$: Sometimes, $T$ could mean a random dataset sampled from $P$ (random). At other times, $T$ could mean a given realization of $T$ (fixed).
> >
> > Now, let's first describe the system setting in our SAFARI algorithm again to make sure that we are all on the same page. Here, the server-side dataset $T$ is uniformly and independently sampled from the data distribution of the entire population $P$. Next, we will specifically discuss where the confusion of expectation comes from.
> >
> > First, in the reviewer's latest comment, the expectation in $\mathbb{E}[ \nabla \hat{F}(x^k)]$ is conditioned on *a realization of the random dataset $T$*, i.e., $T$ is given and fixed. This is reflected in the reviewer's comment *"...You sample $\xi_i^0$ only once in the beginning of algorithm...."* Thus, the expectation in the reviewer's comment is equal to the sample mean in this realization of $T$, i.e., $\mathbb{E}[ \nabla \hat{F}(x^k)] = \mathbb{E} [\frac{1}{k}\sum_i \nabla F(x^k,\xi_i^0)]$. In this case, the reviewer is *correct* that $x^k$ is random and $\mathbb{E}[ \nabla \hat{F}(x^k)] \ne \mathbb{E}[\nabla F(x^k)]$ in general.
> >
> > However, in our analysis and dicussion in the paper, the notation $\mathbb{E}[ \nabla \hat{F}(x^k)]$ represents the *full expectation* that is averaged over i) all randomness of sampling $T$ from $P$; ii) all randomness of the population distribution $P$; and iii) all random trajectories of $x^i$-iterates up to iteration $k$. Note that since $T$ is uniformly sampled from $P$, the full expectation over the joint sampling and population distribution remains equal to $P$. Thus, it is clear that $\mathbb{E}[ \nabla \hat{F}(x^k)] = \mathbb{E}[\nabla F(x^k)]$.
> >
> > We hope the above clarification could clear the reviewer's doubt. Please let us know if there is still any unresolved question. We will continue to try our best to answer. Thanks again for your comments and questions, which definitely improve the clarity of this paper!

---

> > > ### Comment · Reviewer_dvBg · 2022-11-18
> > > **Response**
> > >
> > > Thank you! I still don't agree with you.
> > >
> > > So you agree that conditional expectation $E_t[\nabla F(x^t; \xi^0)] = \frac{1}{k} \sum_i \nabla F(x^t; \xi^0_i)$ is indeed biased, right? But In the proof (page 15, proof Theorem 4, closer to the end of the page), you use the fact that it is unbiased, i.e., $E_t[\nabla F(x^t; \xi^0)] = F(x^t).$ That contradicts your arguments.

---

> > > > ### Comment · Reviewer_dvBg · 2022-11-18
> > > > **Clarrification**
> > > >
> > > > If you still don't agree, then please provide proof of your claims with good notations and clarifications. For instance, in Algorithm 2, is it not clear what is $\xi^0_{t,k}$ (it is only clear from your comments in the rebuttal). What is $E_t$ in the proof? Is it conditioned on $T$ or not? It looks like it should be conditioned on $T,$ because otherwise, you won't be able to insert $E_t$ in the dot product of the first line of the proof of Theorem 4.

---

> > > > > ### Author Response · Authors · 2022-11-19
> > > > > **Response**
> > > > >
> > > > > **Our Response:** Thanks again for your comments. We believe it is a notation confusion that hinders the reviewer to understand the expectation. We make the following clarifications.
> > > > >
> > > > > 1. On Page 15, $\mathbb{E}_t$ means taking expectation over random dataset $T$ conditioning on $x_t$. Hence, $x_t$ is fixed and not random in here. Also, we want to emphasize that $\mathbb{E}_t[\cdot]$ is over all realizations of random dataset $T$, which is uniformly and independently sampled from $P$, rather than the sample mean of a single realization of $T$ (i.e., a fixed dataset). In other words, the random samples $\xi_k^0$ within $\mathbb{E}_t[\cdot]$ is an i.i.d. data sample from a random dataset $T$, which in turn is uniformly sampled from $P$. This implies that $\xi_k^0$ is drawn from the population distribution $P$. Hence, $\mathbb{E}_t [ \nabla \hat{F}(x^k)] = \mathbb{E}_P [\frac{1}{k}\sum_i \nabla F(x^k,\xi_k^0)] = \nabla F(x^k)$.
> > > > >
> > > > > 2. Also, we want to clarify that the subscript $t$ in $\mathbb{E}[\cdot]$ denotes the communication round index, which is irrelevant to dataset $T$. Then, following the well-known descent lemma, we have
> > > > > $\mathbb{E}_t [F(x\_{t+1})] \leq F(x_t) + \langle \nabla F(x_t), \mathbb{E}_t [x\_{t+1} - x_t] \rangle + \frac{L}{2} \mathbb{E}_t [\| x\_{t+1} - x_t \|^2]$
> > > > > $= F(x_t) + \big< \nabla F(x_t), \eta \mathbb{E}_t \bar{g}_r \big> + \frac{L}{2} \eta^2 \mathbb{E}_t [ \| \bar{g}_r \|^2 ].$ In this case, following from our argument in Point 1 above, we can safely insert the expectation into the dot product as $x_t$ is fixed (the reviewer also agreed earlier that the equality holds when $x_t$ is fixed).
> > > > >
> > > > >
> > > > > 3. We understand the reviewer's argument that $\mathbb{E} [ \nabla \hat{F}(x^k)] = \frac{1}{k}\sum_i \nabla F(x^k,\xi_i^0)$ is a biased estimation of $\nabla F(x^k)$. But this is true when $\xi_i^0$ is sampled from a single realization of $T$ (a fixed dataset $T$ drawn from $P$). That is, there is no randomness in dataset $T$. But this is *not* the setting that we analyze in our proof. In our proof and analysis, $\mathbb{E}_t[\cdot]$ considers two sources of randomness: the first one is to consider all realizations of $T$ that are uniformly and independently sampled from $P$; the second one is for the i.i.d. data sampling from $T$.
> > > > >
> > > > >
> > > > >
> > > > > For notation's clarity, we have revised the notations in our paper as follows:
> > > > > 1. We have replaced all communication round index $t$ by $r$ to avoid confusions with the dataset notation $T$.
> > > > > 2. We have added a detailed definition for  each expectation operation that we use.
> > > > > 3. We have added a clear explanation for the random samples $\xi_{r,k}^0$.
> > > > >
> > > > > These modifications are highlighted in **blue**. Please see our revision (Pages 15 and 16). Please let us know if there is still any unresolved question. We will continue to try our best to answer. Thanks again for your comments and questions, which definitely improve the clarity of this paper!

---

> > > > > > ### Comment · Reviewer_dvBg · 2022-11-19
> > > > > > **Response**
> > > > > >
> > > > > > Unfortunately, you can't do this.
> > > > > >
> > > > > > Let me clarify it again. I think we all agree that we have a dataset $P = \xi_k$ $k = 1, ..., |P|$ and a dataset $T = \xi^0_k$ $k=1,..., |T|$  that is once uniformly sampled from $P.$
> > > > > >
> > > > > > We want to find $\frac{1}{k} \sum_{k} E[\nabla F(x^k, \xi^0_k) |x^{k}]$ - a conditional expectation that is conditioned on a point $x^k.$ While $\xi^0_k$ is indeed a uniform sample of $P,$ the problem here is that ***conditional*** distribution of $\xi^0_k$ is not uniform anymore provided that $x^k$ is fixed.
> > > > > >
> > > > > > In other words, I agree that a probability $P(\xi^0_1 = \xi_ j) = 1 / |P|, j =1,..., |P|$ but, in general, $P(\xi^0_1 = \xi_j | x^{k} = \widehat{x}) \neq P(\xi^0_1 = \xi_k),$ where $\widehat{x}$ is not random vector, because $x^{k}$ depends on $\xi^0_1.$
> > > > > >
> > > > > > The fact that we know $x^{k},$ gives us additional information about sampling $T,$ thus, in general, it is not uniform anymore.
> > > > > >
> > > > > > Imagine that |T| = 1, then for SGD we have
> > > > > >
> > > > > > $x^{1} = x^{0} - \gamma \nabla f(x^0; \xi^0_0).$
> > > > > >
> > > > > > Here the stochastic gradient is unbiased $E[\nabla f(x^0; \xi^0_0)] = \nabla f(x^0).$
> > > > > > Let me consider the next step:
> > > > > >
> > > > > > $x^{2} = x^{1} - \gamma \nabla f(x^1; \xi^0_0).$
> > > > > >
> > > > > > Then in
> > > > > >
> > > > > > $E[\nabla f(x^1; \xi^0_0) | x^1 = \widehat{x}]$
> > > > > >
> > > > > > I know that $x^1 = x^{0} - \gamma \nabla f(x^0; \xi^0_0) = \widehat{x},$ so, in general, the condition gives me extra information, and the conditional expectation is not unbiased anymore.
> > > > > >
> > > > > > -----------------------
> > > > > >
> > > > > > Imagine that you want to find the mean of a standard normal variable $\xi.$ We know that $E[\xi] = 0,$ but $E[\xi | f(\xi)] = E[\xi | f(\xi) = \widehat{x}] \neq 0.$

---

> > > > > > > ### Author Response · Authors · 2022-11-22
> > > > > > > **Response**
> > > > > > >
> > > > > > > **Our Response:** Thanks again for your comments. Again, there remain some confusions about the notation. Let us first repeat and explain the meanings of our notation:
> > > > > > >
> > > > > > > * $P$ is the **distribution** of the global dataset of the entire client population. Note that $P$ is *not* a finite dataset as the reviewer suggested.
> > > > > > > * $T$ is a server-side dataset with each sample unformly and independently drawn from the global dataset, whose distribution is $P$. As a result, each sample $\xi^0$ in $T$ follows the distribution $P$.
> > > > > > > * $\mathbb{E}\_r[\cdot]$ is the conditional expectation conditioned on $x_r$, which is averaged over all realizations of the random dataset $T$ and all $K$-step random trajectories between $x_r$ in round $r$ and $x_{r+1}$ in round $r+1$.
> > > > > > >
> > > > > > > It is true that, once $T$ is sampled out of distribution $P$, $T$ is fixed during a single run of our SAFARI algorithm. **However, our goal in this paper is to evaluate the average performance of an infinite number of runs of our SAFARI algorithm, each of which uses a (fixed) realization of dataset $T$.**
> > > > > > >
> > > > > > > Also, we believe that the reviewer is confused about the **meaning of the index $k$**. In our computation $\frac{1}{K}\sum_{k=0}^{K-1}\nabla F(x_{r,k}^{0}, \xi_{r,k}^{0})$ (see Page 15, 4 lines from the bottom), the index $k$ is the index of **local steps** on the server side starting from $x_{r,k}^{0}$ in round $r$. Thus, $\frac{1}{K}\sum_{k=0}^{K-1}\nabla F(x_{r,k}^{0}, \xi_{r,k}^{0})$ is the average of $K$ local steps at the server. Also, using our new index $r$ for round $r$ (we updated it from $t$ to $r$ to avoid potential confusion with dataset $T$), **what we are conditioning on is $x^r$ (notice that $x_{r,0}^0 =x^r$), not $x^k$** (see our blue text on Page 15)
> > > > > > >
> > > > > > > Therefore, the quantity $\frac{1}{K}\sum_{k=0}^{K-1}\nabla F(x_{r,k}^{0}, \xi_{r,k}^{0})$ (Page 15, 4 lines from bottom) is **not the same as** the "sample mean quantity $\frac{1}{k}\sum_k \mathbb{E}[\nabla F(x^k, \xi_0^k)|x^k]$ averaged over all samples in $T$ conditioned on $x^k$" in the reviewer's mind (reflected in the reviwer's comment *"We want to find $\frac{1}{k}\sum_k \mathbb{E}[\nabla F(x^k, \xi_0^k)|x^k]$ - a conditional expectation that is conditioned on a point $x^k$"*). In here, the reviewer thought that $k$ is the index of samples in one realization of the dataset $T$ (reflected in the reviewer's comment *"a dataset $T=\xi_0^k$, $k=1,\ldots,|T|$"*).
> > > > > > >
> > > > > > > We hope that, from the above, the reviewer can clearly see that our previous conversations were actually talking about two different things. Next, let's explain our proof on Page 15. In fact, the proof strategy we used is quite standard in SGD-type algorithms (see, e.g., the classical finite-time convergence analysis of SGD in [R1]), which contains the following key steps:
> > > > > > >
> > > > > > > * Step 1: Consider in round $r$, given $x_r$, we evaluate the expected next-round function value $\mathbb{E}[F(x_{r+1})]$ computed by the server, which can be upper bounded by the standard descent lemma (a consequence of $L$-smoothness, see the blue text on Page 15). Since dataset $T$ is randomly sampled from distribution $P$, the expectation $\mathbb{E}_r[\cdot]$ conditioned on $x_r$ is averaged all realizations of $T$.
> > > > > > > * Step 2: Analyze and futher bound the $A_1$ and $A_2$ terms in the upper bound of $\mathbb{E}[F(x\_{r+1})]$. Regarding your question in the bounding process of $A_1$, the equality $\mathbb{E}_r[\frac{1}{K}\sum\_{k=0}^{K-1}\nabla F(x\_{r,k}^{0}, \xi\_{r,k}^{0})] = \frac{1}{K} \sum\_{k=0}^{K-1}\nabla F(x\_{r,k}^0)$ follows from the fact that $\mathbb{E}_r[\cdot]$ is averaged over all realizations of $T$ and all $K$-step random trajectories from $x_r$ to $x\_{r+1}$.
> > > > > > > * Step 3: Taking full expectation over all random trajectories of rounds $1,\ldots,R$ and by using a telescoping sum, we arrive at a bound on $R$-round descent. Finally, by rearranging terms in the $R$-round descent bound, we obtain our desired nvergence rate result in terms of the stationarity gap.
> > > > > > >
> > > > > > >
> > > > > > > Again, we understand the reviewer's point for the biased expectation. However, due to the confusion in the index $k$, the reviewer's argument is not relevant to the goal in our proof. Moreover, in our convergence analysis, what we evaluate the **expected performance of the SAFARI algorithmis averaged over all realizations** (i.e., all possible realizations of $T$).
> > > > > > >
> > > > > > > [R1] Saeed Ghadimi and Guanghui Lan, "Stochastic First- and Zeroth-Order Methods for Nonconvex Stochastic Programming," SIAM Journal on Optimization, Vol. 23, Iss. 4, 2013.

---

> > > > > > > > ### Comment · Reviewer_dvBg · 2022-11-22
> > > > > > > > **Response**
> > > > > > > >
> > > > > > > > Yes, I had a typo. Sorry for the confusion.
> > > > > > > > I should have written:
> > > > > > > >
> > > > > > > > "We want to find $E[\nabla F(x^j, \xi^0) |x^{j}]$ - a conditional expectation that is conditioned on a point $x^j.$"
> > > > > > > >
> > > > > > > > ***But it does not change my arguments that the proof of SAFARI has a fatal mistake.***
> > > > > > > >
> > > > > > > > Regarding:
> > > > > > > > *However, our goal in this paper is to evaluate the average performance of an infinite number of runs of our SAFARI algorithm, each of which uses a (fixed) realization of dataset.*
> > > > > > > >
> > > > > > > > ***Please formalize this goal mathematically and provide the corresponding proofs.***
> > > > > > > >
> > > > > > > > *Again, we understand the reviewer's point for the biased expectation. However, due to the confusion in the index $k.$*
> > > > > > > >
> > > > > > > > Again, there is no confusion. I simply used the same index $k$ two times. I'll try to repeat my previous argument with the corrected index.

---

> > > > > > > > ### Comment · Reviewer_dvBg · 2022-11-22
> > > > > > > > **Response (Fixed Typos)**
> > > > > > > >
> > > > > > > > We have a dataset $P = \xi_k$ $k = 1, ..., |P|$ and a dataset $T = \xi^0_k$ $k=1,..., |T|$  that is once uniformly sampled from $P.$
> > > > > > > >
> > > > > > > > We want to find $E[\nabla F(x^k, \xi^0) |x^{k}]$ - a conditional expectation that is conditioned on a point $x^k$ and $\xi^0$ is a random sample from $T.$ While $\xi^0$ is indeed a uniform sample of $P,$ the problem here is that ***conditional*** distribution of $\xi^0$ is not uniform anymore provided that $x^k$ is fixed.
> > > > > > > >
> > > > > > > > In other words, I agree that a probability $P(\xi^0 = \xi_ j) = \sum_{m=1}^{T} P(\xi^0 = \xi_ j|\xi^0 = \xi^0_m)P(\xi^0 = \xi^0_m) = \sum_{m=1}^{T} P(\xi^0_m= \xi_ j)\frac{1}{|T|}$
> > > > > > > > $= \sum_{m=1}^{T} \frac{1}{|P|}\frac{1}{|T|} = 1 / |P|, j =1,..., |P|$ but, in general, $P(\xi^0 = \xi_j | x^{k} = \widehat{x}) \neq P(\xi^0_1 = \xi_k),$ where $\widehat{x}$ is not random vector, because $x^{k}$ depends on $\xi^0_k.$
> > > > > > > >
> > > > > > > > The fact that we know $x^{k},$ gives us additional information about sampling $T,$ thus, in general, it is not uniform anymore.

---

> > > > > > > > ### Comment · Reviewer_dvBg · 2022-11-23
> > > > > > > > **Performance of an infinite number of runs of our SAFARI algorithm**
> > > > > > > >
> > > > > > > > *However, our goal in this paper is to evaluate the average performance of an infinite number of runs of our SAFARI algorithm, each of which uses a (fixed) realization of dataset T*
> > > > > > > >
> > > > > > > > This is something that confuses me. In optimization methods, we analyze a single random run: an algorithm generates a random sequence of iterates, and then we analyze its statistical properties.
> > > > > > > >
> > > > > > > > But you want to analyze the infinite number of random sequences. This is something unusual. I suggest the authors clarify in the paper what they mean by that.

---

> > > > > > > > > ### Author Response · Authors · 2022-11-27
> > > > > > > > > **Response**
> > > > > > > > >
> > > > > > > > > **Our Response:** Thanks again for your comments. Let's clarify one more confusion in the reviewer's latest comment: we did *not* analyze the performance of our SAFARI algorithm over "an infinite number of random sequence."" Note that although $T$ a randomly chosen dataset, it is fixed in each run of SAFARI. Thus, $T$ is not an infinite random sequence.
> > > > > > > > >
> > > > > > > > > Again, our interest in this paper is to evaluate **"given $R$ communication rounds, how close the average stationarity gap $\frac{1}{R}\sum_{r=0}^{R-1} \mathbb{E}[ \| \nabla F(x_r) \|^2 ]$ to zero. Here, the expected stationarity gap is averaged over *an infinite number of trials* of our SAFARI algorithm, where each trial uses a fixed dataset $T$ that is sampled from distribution $P$.** Note that the performance metric of average stationarity gap is widely used in the literature of stochastic optimization for machine learning to measure finite-time convergence rate (see, e.g., the classic paper [R1]).
> > > > > > > > >
> > > > > > > > > To understand what we meant by saying "average stationarity gap after $R$ rounds over an infinite number of trials of the SAFARI algorithm", let us revisit the root of stochastic optimization for machine learning. Consider the standard formulation of stochastic optimization problem $\min_{x\in\mathbb{R}^n} \mathbb{E}\_{\xi\sim \mathcal{D}}[f(x,\xi)]$, where $f(x)$ is non-convex in $x$ in general. Thus, finding a near-stationary solution after a finite number of iterations $R$ is typically accepted [R1]. Clearly, the standard approach to solve this problem is the stochastic gradient descent (SGD) method. To analyze the finite-time convergence rate of SGD, in the classic work [R1] (and many other follow-up work), the key performance metric of interest is to bound the stationarity gap $\frac{1}{R}\sum\_{r=0}^{R-1} \mathbb{E}[ \| \nabla F(x_r) \|^2 ]$. Here, the expectation $\mathbb{E}[\cdot]$ is needed since the trajectory $x_{[1:R]}$ under SGD is a random  sequence (randomness comes from the mini-batch sampling in each iteration). Therefore, one needs to run SGD an infinite number of trials over all possible trajectories $x_{[1:R]}$ to obtain the average (expected) performance.
> > > > > > > > >
> > > > > > > > > Now, the performance evaluation of our SAFARI algorithm also follows the same logic as above. Compared to SGD-based approaches for conventional federated learning (FL), the new element in our SAFARI algorithm (for SA-FL) is clearly the server-side dataset $T$, which is randomly sampled from distribution $P$. To evaluate the finite-time stationarity gap $\frac{1}{R}\sum_{r=0}^{R-1} \mathbb{E}[ \| \nabla F(x_r) \|^2 ]$ for our SAFARI algorithm, following the *law of iterated expectation* and the linearity of expectation, we can rewrite the expectation in the finite-time stationarity gap as follows: $\mathbb{E}\_{T\sim P}[\frac{1}{R}\sum_{r=0}^{R-1} \mathbb{E}\_{x_{[1:R]}}[ \| \nabla F(x_r) \|^2 | T ]]$, i.e., the outer expectation is averaged over all realizations of $T$ and the inner expectation is average all random trajectories $x_{[1:R]}$ under a given server-side dataset $T$. As a result, to evaluate the outer expectation over $T$, one needs to run our SAFARI algorithm an infinite number of trials, where in each trial, we need to average over all random trajectories $x_{[1:R]}$ under a fixed server-side dataset $T$.
> > > > > > > > >
> > > > > > > > > [R1] Saeed Ghadimi and Guanghui Lan, "Stochastic First- and Zeroth-Order Methods for Nonconvex Stochastic Programming," SIAM Journal on Optimization, Vol. 23, Iss. 4, 2013.

---

> > > > > > > > > > ### Comment · Reviewer_dvBg · 2022-11-28
> > > > > > > > > > **Response**
> > > > > > > > > >
> > > > > > > > > > Thank you for the comment!
> > > > > > > > > >
> > > > > > > > > > The discussion and arguments of this thread went too far from the paper. I encourage the authors to add all these explanations and arguments. All sources of randomness and defined conditional expectations should be transparent.

---

### Official Review · Reviewer_fscs · 2022-11-03

**Confidence:** 3
**Clarity, Quality, Novelty And Reproducibility:** The paper is well written.
**Correctness:** 4
**Technical Novelty And Significance:** 3
**Empirical Novelty And Significance:** 3
**Recommendation:** 6

**Strength And Weaknesses:**

The paper provides interesting theoretical insights about heterogeneity and the important challenge of partial participation in FL.

SAFARI converges sublinearly to a neigborhood of a stationary point.
I have a question: in Assumption 3: what is partial at the end? In any case, the bounded gradient assumption is restrictive and there seems to be no way to study linear convergence under strong convexity, for instance, since this is not compatible with this assumption.

**Summary Of The Paper:**

The paper is about dealing with partial participation in federated learning. Indeed, in the heterogeneous setting, there is a drift in the obtained solution if not all clients participate. the authors study this discrepancy and propose new algorithms to mitigate it. They consider the idea of sever-aided federated learning (SA-FL), which is to equip the server with a small auxiliary dataset that approximately mimics the population distribution. They provide new results on SA-FL and propose a new algorithm, called SAFARI, which handles partial participation and is communication-efficient.

**Summary Of The Review:**

The paper provides some new insights on partial participation, which is a timely and important problem in modern distributed learning settings. It is nice to obtain theoretical findings on SA-FL, which was so far heuristic.

---

> ### Author Response · Authors · 2022-11-16
> **Response to Reviewer fscs**
>
> Thank you very much for your review and constructive comments. We appreciate the reviewer's recognition of the significance and the potential impact of our work as well as the valuable suggestion.

---

### Official Review · Reviewer_uhzS · 2022-11-03

**Confidence:** 3
**Correctness:** 3
**Technical Novelty And Significance:** 3
**Empirical Novelty And Significance:** 2
**Recommendation:** 5

**Clarity, Quality, Novelty And Reproducibility:**

Clarity:
This paper is easy to follow, and the key idea is successfully conveyed.

Quality:
The writing quality is good.

Novelty:
The theoretical analysis of conventional FL lower bound and SA-FL PAC-learnability is novel. The proposed algorithm SAFARI is novel but a bit intuitive.

Reproducibility:
The illustration of Alg. 1 is almost clear. Lack of experimental details, and possibly there are some issues in the experimental design (concerns above). No code is provided.

**Strength And Weaknesses:**

Strength:
1. This paper theoretically proves conventional FL under the heterogeneous and arbitrary client participation setting is not PAC-learnable even in the limit of infinitely many data samples and training iterations. This could be a very interesting exploration.
2. Based on conventional FL, SA-FL is theoretically analyzed and shown to be PAC-learnable under a few strong assumptions.
3. SAFARI algorithm is proposed for handling SA-FL and achieves almost the same convergence rate as the state-of-the-art conventional FL algorithm under particular conditions.

Concerns:
1. It's unclear whether the Assumption. 1 (noise condition) holds on real datasets, which may have very large noise. E.g., it would be interesting if some real example could be shown that the noise condition is satisfied.
2. It's unclear whether different training protocols influence performance in SA-FL. Given client-distributed dataset $S_D$ with cardinality $n_S$ and server-persisted dataset $T$ with cardinality $n_T$, whether different training strategies affect the robustness and convergence? More concretely, will the following two strategies lead to the same behavior? a) using $S_D$ as a whole for fixed iterations $T^d>0$ and then consider $T$ only (without future averaging) for the following interactions; b) same as above but after $T^d$, do more averaging. What if $T^d=0$?
3. More comprehensive experiments should be provided. In Tab. 1 and Tab. 2 in the main content, only the "improvement" of SAFARI over FedAvg and SGD is provided; I'm curious about the absolute performance. Are SAFARI, FedAvg, and SGD fine-tuned? How are the hyper-parameters selected? Besides, I checked Fig. 4 in the appendix, e.g., (d), SAFARI is always no better than SGD; more analysis should be provided.
4. Seversize data size seems to be too large. MNIST and CIFAR10 both have 60K images, and in this paper, 50-1000 data size from the distribution $p$ is selected; it seems the proportion is too large and not representative in practice. I believe SA-FL/FL is similar to the difference between few-shot learning and zero-shot learning, but only 1, 2, or at most 10 samples per class are provided for the latter. I'm concerned with the performance of a smaller server data size, which is more practical and interesting.
5. In Sec. 5, "to simulate data heterogeneity, distribute the data into each client evenly." I'm unsure if each client has the same data distribution in this setting. What if a different and more heterogeneous data-splitting scheme is applied? Since this paper mainly focuses on the heterogeneous data setting, this should be more carefully considered.

Additional comment:
1. Regarding the difference between FL and SA-FL, the authors may illustrate it with zero-shot and few-shot learning, which is almost the same case. This could be an interesting thinking perspective.

**Summary Of The Paper:**

This work theoretically analyzes that conventional Federated Learning (FL), heterogeneous and arbitrary client participation, is not Probably Approximately Correct (PAC) learnable. And then explore the reason and the fact that the server-aided federated learning (SA-FL) framework uses an auxiliary server dataset to reduce the distribution deviation induced by partial client participation (I will abbreviate it as PP), leading to the PAC-learnability. Besides, an algorithm SAFARI is proposed, which guarantees convergence. Ablation studies on MNIST and CIFAR10 verify the efficiency of the proposed algorithm.

**Summary Of The Review:**

This work is an interesting paper that theoretically explores partial client participation in federated learning. The theories are inspiring, but more comprehensive experiments should be conducted. I would like to increase my score if the above concerns are resolved.

---

> ### Author Response · Authors · 2022-11-16
> **Response to Reviewer uhzS [1/3]**
>
> Thank you very much for your review and constructive comments, which helped us significantly improve the quality of this paper. In this revision, we have carefully revised our paper based on your comments, questions, and suggestions. The detailed point-by-point responses are as follows:
>
> > **Your Comment 1:** It's unclear whether the Assumption. 1 (noise condition) holds on real datasets, which may have very large noise. E.g., it would be interesting if some real example could be shown that the noise condition is satisfied.
>
> **Our Response:** Thanks for your comments. The noise condition, known as Bernstein noise condition, is a standard and widely used assumption in statistical learning with noise, as stated in the paper. Note that  the Bernstein noise condition is characterized by not only the hypothesis space and the underlying dataset distribution, but also the sampling procedure over the dataset. Thus, whether or not the Bernstein noise condition holds on real datasets is not a straightforward question due its dependence on these three factors.
>
> Interetsingly, one special case where the Bernstein noise condition always holds is in the *over-parameterized regime* (i.e., deep learning models) with the so-called "interpolation effect." Specifically, in the over-parameterized regime with interpolation effect, there exists a target hypothesis (model) $f$, such that for all i.i.d. data $(x, y)$ sampled from distribution $P$, we have $f(x) = y$. In other words, the model $f$ could fit every data $(x, y)$ in the dataset. In this special case, it follows that $\mathbb{P}(Y = f(X)|X) = 1$. It can be readily verified that the Bernstein noise condition always holds with $\alpha = \beta = 1$ in this special case.
>
> For general machine learning models (including non-over-parameterized models where the interpolation effect does not occur), to our knowledge, studies on how well the Bernstein noise condition is satisfied on real datasets remain very limited in the literature. But from a fair comparison perspective with other generalization peroformance results in the literature, we still adopt this widely used Berstein noise condition in this paper. Also, it is worth pointing out that the Beinstein noise condition is considered quite general, which unifies and extends many other noise conditions, such as the Tsybakov noise condition [a], $\beta$-bounded noise condition [b], etc.
>
> [a]. Massart, Pascal, and Élodie Nédélec. "Risk bounds for statistical learning." The Annals of Statistics 34.5 (2006): 2326-2366.
>
> [b]. Hanneke, Steve. "Refined error bounds for several learning algorithms." The Journal of Machine Learning Research 17.1 (2016): 4667-4721.

---

> > ### Author Response · Authors · 2022-11-16
> > **Response to Reviewer uhzS [2/3]**
> >
> > > **Your Comment 2:** It's unclear whether different training protocols influence performance in SA-FL. Given client-distributed dataset $S_D$ with cardinality $n_s$ and server-persisted dataset $T$ with cardinality $n_T$, whether different training strategies affect the robustness and convergence? More concretely, will the following two strategies lead to the same behavior? a) using $S_D$ as a whole for fixed iterations $T^d > 0$ and then consider T only (without future averaging) for the following interactions; b) same as above but after $T^d$, do more averaging. What if $T^d=0$?
> >
> > **Our Response:** Thanks for your comments. We organized our responses to this questions in to two parts:
> >
> > * *Traning:* Clealy, different training protocols (i.e., algorithms) could have a significant impact on the training convergence performance of SA-FL, which will in turn affects both sample and communication complexities in SA-FL training. In fact, in the second half of our paper, we show that our proposed SAFARI algorithm enjoys a good training convergence performance under SA-FL. Regarding your proposed protocols/algorithms, interestingly, they can all be viewed as special cases of our SAFARI algorithm. Specifically, Protocol a) corresponds to setting $c_t=0$ in SAFARI for $t\geq T^d$, and Protcol b) corresponds to allowing $c_t > 0$ in SAFARI for $t\geq T^d$. Note that, with an appropriately chosen $T^d$-value, both protocols can satisfy the condition $\sum_{t\in[T]} c_t^2 < \infty$. Thus, our results in Theorem 4 are also applicable for these two protocols. Also, the case with $T^d=0$ can also be captured by setting $c_t=0$, $\forall t$. Plugging this setting into Theorem 4 would yield the convergence performance of the setting with $T^d=0$. Note that, in this case, the convergence error bound is smaller due to $c_t=0$, $\forall t$. This makes intuitive sense because the server-side dataset is typically much smaller, thus inducing a faster convergence.
> >
> > * *Generalization and Robustness:* Different training protocols will also have a significant impact on the generalization performance of the trained model under SA-FL. However, regarding the generalization performance in this paper, we are more interested in the *"fundamental limit"* of SA-FL, which is typically characterized by the PAC (probably approximately correct) learnability framework in statistical learning theory. In the PAC learnablity framework, one is more concerned with the worst-case generalization error under *all possible training algorithms* in a particular learning regime (SA-FL in this case). If the worst-case generalization error bound of all possible training algorithms under a learning paradigm can be made arbitrarily small (typically in the sense of "with high probability") as the number of samples increases, this learning paradigm is termed "PAC learnable." In other words, the PAC framework informs *"how a learning paradigm can generalize under all possible algorithms."* Because of this, our PAC-learnabilty results in Theorems 1 and 2 cannot be used to characterize the generalization performance of a given training protocol. To show the generalization performance of a given training algorithm (such as those you suggested or our SAFARI algorithm), other more appropriate analytical frameworks could be employed (e.g., the algorithmic stability framework in [c]), but this is somewhat beyond the scope of our current paper and deserves an independent paper. We agree that all these are very interesting open problems in SA-FL, which warrants in-depth investigations in our future studies. We thank the reviewer for pointing these directions!
> >
> > [c] Hardt, M., Recht, B., and Singer, Y. Train faster, generalize better: Stability
> > of stochastic gradient descent. In International Conference on Machine Learning
> > (2016), pp. 1225–1234.
> >
> > > **Your Comment 3:** More comprehensive experiments should be provided. In Tab. 1 and Tab. 2 in the main content, only the "improvement" of SAFARI over FedAvg and SGD is provided; I'm curious about the absolute performance. Are SAFARI, FedAvg, and SGD fine-tuned? How are the hyper-parameters selected?
> >
> > **Our Response:** Thanks for your comments. In terms of "absolute performance," with partial client participation $s=4$, the best test accuracy of SAFARI is above 92% in the i.i.d. case for all server data sizes. With non-i.i.d. index $p=1$ and server data size being 50, SAFARI and SGD in centralized learning have test accuracy 69.4% and 65.7%, respectively. For parameter-tuning, SAFARI, FedAvg and SGD in centralized learning all share the same parameters: the learning rate is 0.1, and the local epoch is 1; the batch size of the client is 64, and the total number of communication rounds is 150.

---

> > > ### Author Response · Authors · 2022-11-16
> > > **Response to Reviewer uhzS [3/3]**
> > >
> > > >**Your Comment 4:** Seversize data size seems to be too large. MNIST and CIFAR10 both have 60K images, and in this paper, 50-1000 data size from the distribution  is selected; it seems the proportion is too large and not representative in practice. I believe SA-FL/FL is similar to the difference between few-shot learning and zero-shot learning, but only 1, 2, or at most 10 samples per class are provided for the latter. I'm concerned with the performance of a smaller server data size, which is more practical and interesting.
> > >
> > > **Our Response:** Thanks for your comments. First, we want to clarify that the relationship between FL and SA-FL is completely *different* from that between few-shot learning and zero-shot learning. Regarding FL and SA-FL, the use of server-side dataset in SA-FL is to help mitigate the bias induced by partial client participation in conventional FL. Both SA-FL and FL are trying to learing the *same* model. Although a small server-side dataset is often preferred, minimizing the size of server-side dataset is often not the biggest goal in SA-FL. In stark contrast, the difference in the number of additional samples in few-shot and zero-shot learning will have different impacts on adapting some pre-trained model to other downstream tasks, i.e., both few-shot and zero-shot learning are focused on transferring from an existing model to new models. In this case, the efficiency of few-shot learning compared to zero-shot learning is clearly characterized by the number of additional samples.
> > >
> > > Regarding the difference between SA-FL and conventional FL, we want to further emphasize that the key design concern is not soley on the size of server-side dataset (as mentioned above), but more importantly, the data distribution of the participating clients. Take an FL system with the MNIST dataset as an example. Consider an extremely biased partial client participation scenario, where the data from all participated clients only has one label. In this scenario, the server-side dataset is neccesarily to be "sufficiently large" in the sense that it should cover other nine classes of labels. On the other hand, in an ideal scenarios with more uniform client participation, the server SA-FL only needs to have a small number of data or even no data. The size of server-side dataset in SA-FL depends on both severity of partial client participation and the learning error tolerance, which is also reflected our theoretical analysis that the server's data (sampled from $P$) should be adjusted accordingly based on $D$ to make sure $Q = P + D \approx P$.
> > >
> > > Lastly, we would like to clarify the "position" of our paper: The goal of this paper is *not* to promote the use of SA-FL. In fact, as discussed in the Introduciton section, SA-FL has been proposd and used by many researchers in practical FL systems, which has found empirical successes in mitigating partial client participation. However, the theoretical understanding of SA-FL remains rather limited so far. Therefore, the goal of this paper is to obtain a deep theoretical understanding of performance of SA-FL. It is true that the size of the server-side dataset could be large, as you rightfully pointed out and we discussed/explained above. Actually, providing such theoretical insights and revealing the above limitations of SA-FL (in server-side dataset size) are exactly the contributions in this paper.
> > >
> > > >**Your Comment 5:** In Sec. 5, "to simulate data heterogeneity, distribute the data into each client evenly." I'm unsure if each client has the same data distribution in this setting. What if a different and more heterogeneous data-splitting scheme is applied? Since this paper mainly focuses on the heterogeneous data setting, this should be more carefully considered.
> > >
> > > **Our Response:** Thanks for your comments. It appears that there is a misunderstanding here. "Evenly" just means that each client has the same number of local data samples, but the distribution at each client could be completely different, since each client holds a different class of labels, i.e., non-i.i.d. dataset. The data heterogeneity is indexed by the $p$-value in our paper, which represents the number of label classes at each client. This is a standard way of constructing non-i.i.d. dataset in the literature (see, e.g., (McMahan et al., 2017; Yang et al., 2021b; Li et al., 2020b) in the paper's refernce list). In our experiments, we consider different heterogeneous data settings ($p=1,2,5,10$). We have modified this sentence in the revision to avoid such a confusion.

---

### Official Review · Reviewer_m3ZD · 2022-11-04

**Confidence:** 4
**Correctness:** 2
**Technical Novelty And Significance:** 2
**Empirical Novelty And Significance:** 2
**Recommendation:** 5

**Clarity, Quality, Novelty And Reproducibility:**

**Clarity**:

The structure of this paper is overall easy to follow. However, there are some confusing statements.

- In Definition 1, there is only one argument of $\mathcal{R}_\mathcal{D}(\cdot)$. However, $\mathcal{R}_P(\cdot,\cdot)$ appears in the last line of Theorem 1.
- Why state "$(\alpha,\beta)$-Positively-Related" as a definition instead of an assumption? It is actually a necessary assumption in Theorem 2.
- The definition of $\mathcal{R}$ in the proof of Theorem 1 is not consistent with that in Definition 1.
- Missing reference on Page 17 of the appendix.

**Reproducibility**:

No submitted code.

**Strength And Weaknesses:**

**Strength**:
- This paper provides theoretical guarantees for a Federated Learning framework that has been adopted in practice.
- The ablation study in Section 5 is insightful.

**Weakness**:

I have several concerns about the theoretical results of this paper.

Major concerns:
- In the paragraph below Theorem 2, it says "Note that when $\beta\geq 1$, the first term in Eq. (1) dominates." This is not true unless $n_T + n_S > d_{\mathcal{H}}$. However, we cannot expect $n_T + n_S > d_{\mathcal{H}}$ to hold for FL with deep neural networks. For example, Bartlett et al. [1] show that $d_\mathcal{H} \geq \Omega(WL\log(W/L))$ for ReLU neural networks, where $W$ and $L$ are numbers of parameters and layers, respectively. If $n_T + n_S \leq d_{\mathcal{H}}$, the bound in Theorem 2 becomes vacuous. This should be explicitly mentioned.
- The assumption on "$(\alpha,\beta)$-Positively-Related" does not seem to be mild/natural. The authors did not justify it or give any example to explain when this condition might hold.
- As far as I can tell, SA-FL degenerates into FL when $n_T = 0$. If we plug $n_T = 0$ into Theorem 2, the result shows that FL is also PAC-learnable under the same assumptions. This seems to contradict the main story of this paper.
- The authors mention that "For example, it is shown in (Yang et al., 2021a) that more than 30% of clients never participate in FL, while only 30% of the clients contribute to 81% of the total computation even if the server uniformly samples the clients.'' However, this paper does not characterize the generalization bounds for participating and nonparticipating clients separately as suggested in [2].
- To ensure convergence, the proposed algorithm SAFARI needs $c_t$ decreases to 0 quickly enough (or even setting $c_t= 0$), which implies that the contribution of the local model updates diminishes to (or is) zero and the algorithm approaches (or is) the centralized training on the auxiliary data. Then, it is not surprising that the convergence guarantee does not depend on heterogeneity.

Minor concerns:
- There is a $K^2\eta^3$ term in Theorem 4. If we choose $\eta = \frac{1}{\sqrt{KT}}$, the term becomes $K^2 \frac{1}{K^{3/2}T^{3/2}} = \frac{K^{1/2}}{T^{3/2}}$, which does not appear in the convergence rate shown in Corollary 1.
- Why the error bars are not shown in Figure 2?
- Only comparing SAFARI to FedAvg (LocalSGD) is not enough. More modern baselines such as SCAFFOLD (Karimireddy et al. 2019), FedProx (Li et al. 2018), and Nastya (Malinovsky et al. 2022) are needed to be compared with for showing the effectiveness of the proposed algorithm.

[1] Bartlett, Peter L., Nick Harvey, Christopher Liaw, and Abbas Mehrabian. "Nearly-tight VC-dimension and pseudodimension bounds for piecewise linear neural networks." The Journal of Machine Learning Research 20, no. 1 (2019): 2285-2301.

[2] Yuan, Honglin, Warren Richard Morningstar, Lin Ning, and Karan Singhal. "What Do We Mean by Generalization in Federated Learning?." In International Conference on Learning Representations. 2021.





**Summary Of The Paper:**

> After the rebuttal: Some of my concerns have been resolved. However, the optimization theory still does not explain why $c_t = 0$ is not the optimal one. Since no theory shows that the generalization error of SA-FL is better than centralized learning. The whole story does not explain the empirical results.

In this paper, the authors study the PAC-learnability of Federated Learning (FL) with or without an auxiliary dataset on the central server. In the regime of FL with an auxiliary dataset on the central server (called SA-FL), they propose a new algorithm called SAFARI in which the central server computes the next round's initial model by both the aggregated model updates from local servers and the model update based on its own dataset.

**Summary Of The Review:**

This paper is the first to establish some theoretical guarantees of a practically used framework called server-aided federated learning (SA-FL). However, the concerns in the "Strength And Weaknesses" section make me unsure about the contributions of this paper.

---

> ### Author Response · Authors · 2022-11-16
> **Response to Reviewer m3ZD [1/4]**
>
> Thank you very much for your review and constructive comments, which helped us significantly improve the quality of this paper. In this revision, we have carefully revised our paper based on your comments, questions, and suggestions. The detailed point-by-point responses are as follows:
>
> > **Your Comment 1:** In the paragraph below Theorem 2, it says "Note that when $\beta \geq 1$, the first term in Eq. (1) dominates." This is not true unless $n_T + n_S > d_{\mathcal{H}}$. However, we cannot expect $n_T + n_S > d_{\mathcal{H}}$ to hold for FL with deep neural networks. For example, Bartlett et al. [1] show that $d_{\mathcal{H}} > \Omega(WLlog(W/L))$ for ReLU neural networks, where $W$ and $L$ are numbers of parameters and layers, respectively. If $n_T + n_S < d_{\mathcal{H}}$, the bound in Theorem 2 becomes vacuous. This should be explicitly mentioned.
>
> **Our Response:** Thanks for your constructive comments. It is true that our PAC generalization error bound in Theorem 2 becomes vacuous in the over-parameterized regime with $n_T + n_S \leq d_{\mathcal{H}}$. However, in what follows, we would like to point out **three reasons** to justify why $n_T + n_S > d_{\mathcal{H}}$ is highly relevant in federated learning (FL) and why our result in Theorem 2 is still useful and interesting:
>
> 1) Consider the so-called *cross-device FL* setting (by far the most prevalent FL model), which is typically deployed over edge networks with a massive number of clients (could be up to $10^{10}$, see Ref. [a]). In such edge networks, clients are typically implemented on mobile devices with limited computation, storage, and communication capabilities. This implies that training big models in the over-parameterized regime over cross-device FL is difficult. In other words, training regular non-deep models is often more relevant in FL. As a result, in generalization analysis for FL, it is safe to assume that $n_T + n_S > d_{\mathcal{H}}$ holds in cross-device FL with reasonably large datasets on the client side.
>
>
> 2) It is worth noting that, in the over-parameterized regime, the looseness of the VC-dimension-based PAC bound in Theorem 2 is *not* due to our proof and analysis. Rather, this deficiency is a technical limitation of the classical VC-dimension-based PAC analytical framework, which is well-known for not being able to explain the good generalization performance of over-parameterized models. This limitation has motivated many recent new advances in statistical learning theory that go beyond the conventional wisdom of VC-dimension-based PAC analysis to explain the generalization of deep learning (e.g., double-descent theory). However, due to the practical relevance of regular non-deep models in FL (as pointed out the previous bullet), we still adopt the classical VC-dimension-based PAC framework in this paper. We agree with the reviewer that it is a very interesting topic to develop new theories for the generalization performance of deep models trained under FL, although this is beyond the scope of the current paper. We will consider this topic in our future work and thank you for this suggestion!
>
> 3) We would like to further point out that, in the context of classical statistical learning theory based on VC-dimension and PAC analysis, our result in Theorem 2 actually *matches* the classic lower bound in terms of the dependence of $\frac{d_{\mathcal{H}}}{m}$ (see, e.g., Thm 3.20 in textbook [b] and Thm 1 in transfer learning [c]), where $d_{\mathcal{H}}$ is VC-dimension and $m$ is the number of data samples. Note that this dependence of $\frac{d_{\mathcal{H}}}{m}$ is not improvable. This implies that our result in Theorem 2 is order-optimal in the sense of $\frac{d_{\mathcal{H}}}{m}$-dependence and aligns with the state-of-the-art results in VC-dimension-based PAC learning analysis.
>
> [a]. Kairouz, Peter, et al. "Advances and open problems in federated learning." Foundations and Trends® in Machine Learning 14.1–2 (2021): 1-210.
>
> [b]. Mohri, Mehryar, Afshin Rostamizadeh, and Ameet Talwalkar. Foundations of machine learning. MIT press, 2018.
>
> [c]. Hanneke, Steve, and Samory Kpotufe. "On the value of target data in transfer learning." Advances in Neural Information Processing Systems 32 (2019).

---

> > ### Author Response · Authors · 2022-11-16
> > **Response to Reviewer m3ZD [2/4]**
> >
> > > **Your Commment 2:** The assumption on "$(\alpha,\beta)$-Positively-Related" does not seem to be mild/natural. The authors did not justify it or give any example to explain when this condition might hold.
> >
> > **Our Response:** Thanks for your comments. Here, we would further explain the intuition of the "$(\alpha,\beta)$-Positively-Related" condition and justify why it is both "mild" and "natural."
> >
> > * The "Natural" Aspect: The "$(\alpha,\beta)$-Positively-Related" condition is directly motivated by and can be viewed as a generalization of the noise condition in Assumption 1, which has been widely used in the literature (see (Massart & Nédélec, 2006; Koltchinskii, 2006; Hanneke, 2016) in our paper's references). Note that the traditional Assumption 1 characterizes, under a single distribution $P$, the gap of a hypothesis $h$ with respect to the optimal hypothesis $h^*$ by using an $(\alpha,\beta)$-positively-related condition of the excess error $\epsilon_P(h)$. Motivated by this mathematical construct in Assumption 1, we propose in this paper to generalize this condition and characterize the gap between the excess errors of *two* distributions $P$ and $Q$ by using a similar $(\alpha,\beta)$-positively-related condition. In fact, we consdier this $(\alpha,\beta)$-positively-related condition between two distributions a new contribution to the literature and could be of independent interest.
> >
> > * The "Mild" Aspect: To see why the $(\alpha,\beta)$-positively-related condition is a mild condition, let's first consider the following "one-dimentional" example for simplicity: Let $\mathcal{H}$ be the class of hypotheses defined on the real line: $\{ h_t = t, t \in R \}$, and let two uniform distributions be $P := \mathcal{U}[a, b]$ and $Q := \mathcal{U}[a', b']$. Due to the partial client sampling in FL, the support of $Q$ is a subset of that of $P$, i.e., $a \leq a' \leq b' \leq b$. Denote the target hypothesis $t^* \in [a', b']$. Then, for any hypothesis $h_t$ with threshold $t$, we have $\epsilon_P(h_t) = \frac{| t - t^* |}{b - a}$ and $\epsilon_Q(h_t) = \frac{| t - t^* |}{b' - a'}$. That is, our "$(\alpha,\beta)$-Positively-Related" holds for $\alpha = 1 - \frac{b' - a'}{b - a}$ and $\beta = 1$. Note that the above "one-dimensional" example could be extended to general high-dimensional cases as follows. Intuitively, the difference of excess errors of $P$ and $Q$ (i.e., $| \epsilon_P(h) -  \epsilon_Q(h) |$) is a function in the form of $\int_S |Q_X - P_X|dS$ for a common support domain $S \subset supp(Q)$. Thus, the "$(\alpha,\beta)$-Positively-Related" condition can be written as $| \int_S Q_X dS - \int_S P_X dS | \leq \alpha (\int_S Q_X)^{\beta}$. If distribution $Q$ has more probability mass over $S$ than distribution $P$, choosing $\beta = 1$ and $\alpha$ to be a sufficiently large constant clearly satisfies the $(\alpha,\beta)$-positively-related condition. Otherwise, letting $\beta \rightarrow 0$ and choosing $\alpha$ to be a sufficeintly large constant satisfies the "$(\alpha,\beta)$-Positively-Related" condition in most cases.
> >
> > [a]. Ben-David, Shai, et al. "A theory of learning from different domains." Machine learning 79.1 (2010): 151-175.
> >
> > [b]. Mohri, Mehryar, and Andres Muñoz Medina. "New analysis and algorithm for learning with drifting distributions." International Conference on Algorithmic Learning Theory. Springer, Berlin, Heidelberg, 2012.

---

> > > ### Author Response · Authors · 2022-11-16
> > > **Response to Reviewer m3ZD [3/4]**
> > >
> > > > **Your Comment 3:** As far as I can tell, SA-FL degenerates into FL when $n_T=0$. If we plug $n_T = 0$ into Theorem 2, the result shows that FL is also PAC-learnable under the same assumptions. This seems to contradict the main story of this paper.
> > >
> > > **Our Response:** Thanks for your comments. It appears that there is some confusion and misunderstanding on our results in Theorems 1 and 2. Here, we want to clarify and emphasize that both Theorems 1 and 2 are all *worst-case* results. Specifically, in the worst case, when setting $n_T=0$, it leads to $Q = D$. As a result, SA-FL degenerates into FL with $\beta = 0$ in the worst case. This yields a non-vanishing $\mathcal{O}(1)$ term in Theorem 2's generalization bound, which is *consistent* with the *worst-case* non-PAC-learnable result of FL with partial client participation in Theorem 1. That is, when $n_T = 0$, FL is not PAC-learnable in the worst case under partial client participation.
> > >
> > > Also, we want to take this opportunity to clarify that the non-PAC-learnable result in Theorem 1 is a *worst-case* result, and it does *not* necessarily mean that every FL instance with partial client participation is not PAC-learnable. Nonetheless, our worst-case "non-PAC-leanable" result in Theorem 1 explains the unsatisfactory empirical performance of FL under partial client participation widely observed in the literature (see Page 1, Section I).
> > >
> > > Lastly, we want to again emphasize that, as mentioned in Section I, a popular apporach to mitigate the impact of partial client participation in FL is the *SA-FL* apporach, which has been widely adopted by many researchers in practical FL systems. However, the theoretical justification of SA-FL remains unclear. Our PAC generalization bound in Theorem 2 provides such theoretical justifications SA-FL and advances our understanding on the use of SA-FL.
> > >
> > > > **Your Comment 4:** The authors mention that "For example, it is shown in (Yang et al., 2021a) that more than 30% of clients never participate in FL, while only 30% of the clients contribute to 81% of the total computation even if the server uniformly samples the clients.'' However, this paper does not characterize the generalization bounds for participating and nonparticipating clients separately as suggested in [2].
> > >
> > > **Our Response:** Thanks for your comments and the pointer to Ref. [2]. Actually, we are aware of the new and interesting generalization framework proposed in Ref. [2], which seperates the out-of-sample gap and participation gap. However, in this paper, our focus is on whether we can characterize the theoretical generalization performance of the (empirically successful) SA-FL framework through the classical lens of PAC-learnability. The rationale of our approach is that PAC-learnablity is a more established notion in the areas of statistical learning, which enables us to better understand and compare conventional FL, SA-FL, and other learning paradigms. Towards this end, not only does our theoretical analysis establish the PAC-learnability for SA-FL, our approach mentioned above also revelas an interesting result that SA-FL enjoys a better generalization error bound than that of domain adaption.
> > >
> > > On the other hand, we do agree that analyzing the generalization performance of conventional FL and SA-FL using the new framework in Ref. [2] is a very interesting topic. To pursue this direction, we believe that some additional assumptions and modellings about the client participations could be needed if we want to characterize generalization bounds for participating and nonparticipating clients separately. We believe these seperated generalization bounds are of theoretical importance in quantifying client diversity and measuring incentives, which warrants an independent and dedicated paper to this topic in our future studies. Again, we want to thank the reviewer for pointing out this interesting direction!

---

> > > > ### Author Response · Authors · 2022-11-16
> > > > **Response to Reviewer m3ZD [4/4]**
> > > >
> > > > > **Your Comment 5:** To ensure convergence, the proposed algorithm SAFARI needs $c_t$ decreases to 0 quickly enough (or even setting $c_t=0$), which implies that the contribution of the local model updates diminishes to (or is) zero and the algorithm approaches (or is) the centralized training on the auxiliary data. Then, it is not surprising that the convergence guarantee does not depend on heterogeneity.
> > > >
> > > > **Our Response:** Thanks for your comments. It is true that, to ensure convergence, the series $\{c_t\}$ is required to be shrinking, so that $\sum_{t \in [T]} c_t^2 < \infty$ or $\sum_{t \in [T]} c_t^2 = \mathcal{O}(\min\{KT, K^{3/2}T^{1/2}\})$ (i.e., $\sum_{t\in[T]} c_t^2$ has a slow growth, see remarks in Paragraph 1, Page 8). However, we emphasize that it is *not necessary* to require $\{c_t\}$ to be rapidly shrinking (or even setting $c_t = 0$), so as to ensure convergence. Specifically, note that we only have a requirement on $\sum_{t\in[T]} c_t^2$, but we don't have any requirement for $\sum_{t \in [T]} c_t$. In fact, $\{c_t\}$ could possibly be shrinking so slowly that $\sum_{t\in[T]} c_t = \infty$. For example, $\{c_t\}$ could be chosen as the famous divergent harmonic series {$ \{1/t \}$}, which still satisfies our conditions $\sum_{t \in [T]} c_t = \infty, \sum_{t \in [T]} c_t^2 < \infty$.
> > > >
> > > > We would also like to point out that the gradual shrinking mechanism of the $\{c_t\}$ series is actually the beauty of our SAFARI algorithm: $\{c_t\}$ serves as a control knob that carefully balances the uses of client-side and server-side data. Intuitively speaking, in the initial stage of the algorithm when the model parameters are far from convergence, we want to leverging more client-side data to improve parallelism by setting a larger $c_t$-value, while still being guided by the server-side data to avoid convergence error. On the other hand, in the later stage of the algorithm when the model parameters are close to convergence, we take a smaller $c_t$-value to use more server-side data to mitigate the negative impacts of partial client participation that could hurt convergence.
> > > >
> > > > > Minor issues: 1) $K^2 \eta^3$ terms in Thm 4. 2) Definition of $\mathcal{R}$. 3) missing reference on page 17
> > > >
> > > > **Our Response:** Thanks for your comments. We have modified the definitions and fixed the typos accordingly in the revision.

---

### Decision · Program_Chairs · 2023-01-20

**Decision:**

Reject

**Justification For Why Not Higher Score:**

Fatal issues, lack of rigour. Result does not improve on standard approaches.

**Justification For Why Not Lower Score:**

N/A

**Metareview: Summary, Strengths And Weaknesses:**

The authors study the PAC-learnability of Federated Learning (FL) with or without an auxiliary dataset on the central server. In the regime of FL with an auxiliary dataset on the central server (called SA-FL), they propose a new algorithm called SAFARI in which the central server computes the next round's initial model by both the aggregated model updates from local servers and the model update based on its own dataset.

The structure of this paper is easy to follow.

Weaknesses:

- The optimization theory in this paper does not explain why $c_t = 0 is not the optimal choice. This is a major issue since this choice is the one used in standard literature. Since no theory shows that the generalization error of SA-FL is better than centralized learning, the paper story does not explain the empirical results.
- As an extensive and detailed discussion with one of the reviewers showed, the paper contains a large number of issues with notation, clarity and mathematical rigour. These are major issues that require a major revision of the paper.


Unfortunately, any single one of these two issues is enough for me to clearly recommend a rejection.